# Shared Recurrent Memory Transformer for Multi-agent Lifelong Pathfinding

## Abstract

Coordination in decentralized multi-agent reinforcement learning (MARL) necessitates that agents share information about their behavior and intentions. Existing approaches rely on communication protocols with domain or resource constraints or centralized training that poorly scales to large agent populations. We introduce the Shared Recurrent Memory Transformer (SRMT), which enables coordination through unconstrained communication. SRMT provides a global memory workspace where agents broadcast their learned working memory states and query others' memory representations to exchange information and coordinate while maintaining decentralized training and execution. We evaluate SRMT on the Partially Observable Multi-Agent Pathfinding (PO-MAPF) problem, where coordination is vital for optimal path planning and deadlock avoidance. We demonstrate that shared memory enables emergent coordination even when the reward function provides minimal or no guidance. On the specifically constructed Bottleneck task that requires negotiation, SRMT consistently outperforms communicative and memory-augmented baselines, particularly under sparse reward signals, and successfully generalizes to longer corridors unseen during training. On POGEMA maps, SRMT scales with the increasing agents' population and map size, achieving competitive performance with recent MARL, hybrid, and planning-based methods while requiring no domain-specific heuristics. These results demonstrate that a transformer with shared recurrent memory enhances coordination in decentralized multi-agent systems.

## 1 Introduction

Enabling coordination between multiple agents is one of the central challenges in multi-agent reinforcement learning (MARL) (Huh & Mohapatra, 2024), often requiring sophisticated communication protocols and decision-making mechanisms (Foerster et al., 2016a; Iqbal & Sha, 2018; Zhang et al., 2018). The coordination problem becomes particularly acute when agents act interdependently under partial observability – one agent's optimal policy becomes subject to predicting and adapting to others' behaviors. Unlike centralized systems, where a global controller can directly assign roles and resolve conflicts, decentralized agents must coordinate through limited information exchange while making independent decisions. Existing solutions to this challenge are devoted to communication protocols that require pre-defined message spaces, and centralized training that harms scaling to large agent populations (Egorov & Shpilman, 2022; Foerster et al., 2016b).

Existing communication methods suffer from several fundamental limitations. They require specific message design (Chu et al., 2020; Kim et al., 2020), consume communication bandwidth (Foerster et al., 2016b; Zhang et al., 2019; Wang et al., 2020), and force agents to perform complex reasoning about others' goals from limited signals. Approaches using the Centralized Training Decentralized Execution (CTDE) (Egorov & Shpilman, 2022) can learn coordination during training but struggle with scalability and often fail to generalize when the number of agents or environment characteristics change during deployment.

We address these limitations by introducing a *shared memory* as a global workspace for agents' coordination. The agents broadcast learned representations of their working memory states, capturing their behavior and environmental understanding into a global memory workspace. Other agents can then query this shared space through an attention mechanism and update their own policies ac-

cordingly. The proposed *Shared Recurrent Memory Transformer (SRMT)* enables rich information exchange without requiring pre-defined constrained communication protocols, while maintaining the scalability benefits of a decentralized training and execution paradigm.

Partially Observable Multi-agent Pathfinding (PO-MAPF) (Stern et al., 2019) is a task of moving a group of agents from their respective start locations to their goal locations without collisions, while each agent observes other agents only locally. The real-world applications that can be directly formulated as MAPF include warehouse management (Sharon et al., 2015; Li et al., 2021b) and swarm control (Li et al., 2021a) among others. PO-MAPF presents a distinctive multi-agent coordination challenge because it requires coordination to be embedded directly into the planning process. It complicates the development of a system that provides useful high-level coordination guidance and integrates it with low-level path planning to produce valid solutions. Coordination failures lead to congestion and deadlocks where agents are unable to progress toward their goals. The NP-hard complexity of MAPF ensures that simple reactive strategies fail, requiring agents to anticipate others' actions and engage in implicit negotiation.

Our key contributions are: (1) We introduce the Shared Recurrent Memory Transformer (SRMT) as a new architectural mechanism for general coordination in decentralized multi-agent systems, providing a *shared "workspace"* jointly written and read by all agents at every step and enabling information exchange without pre-defined communication protocols. (2) We demonstrate that SRMT paired with an actor-critic network and trained with Proximal Policy Optimization (PPO) enables emergent coordination behaviors, as evidenced by learned memory representations that correlate with agents' behavior. (3) We show that SRMT-based coordination generalizes across different environment scales, agent populations, and scenarios, outperforming communication-based and memory-augmented baselines on the Bottleneck navigation task and performing competitively with recent MARL, hybrid, and planning-based algorithms on challenging POGEMA (Skrynnik et al., 2024a) tasks while requiring no domain-specific design choices.

## 2 RELATED WORK

**Coordination Challenges in MARL and MAPF**    The existing methods for enabling coordination under partial observability in general MARL, and in MAPF particularly, can be split according to their training-execution paradigm: (1) centralized approaches where a global controller aggregates information from all agents; (2) decentralized approaches where agents make decisions based solely on local observations; and (3) hybrid approaches combining centralized training and decentralized execution or decentralized methods with networked agents that allow local inter-agent information sharing (Zhang et al., 2021; Hu et al., 2023; Nayak et al., 2023; Agarwal et al., 2019). Centralized MAPF approaches like LaCAM (Okumura, 2023) and RHCR (Li et al., 2021b) achieve optimal coordination by having complete global information, but fail to scale to large agent populations and cannot adapt to deployment scenarios where centralized control is infeasible. Decentralized methods such as IQL (Tan, 1993) avoid coordination overhead but often converge to suboptimal equilibria where agents fail to cooperate effectively. The most promising direction combines centralized training with decentralized execution (CTDE), exemplified by VDN (Sunehag et al., 2018), QMIX (Rashid et al., 2020), and QPLEX (Wang et al., 2021). These methods learn to factorize joint value functions during training while maintaining independent execution. However, they struggle with scalability and generalization when agent populations or environment characteristics change between training and deployment. Another problem is the extreme difficulty of separation of high-level coordination from path planning in MAPF (Chen et al., 2024). To avoid conflicts at each step we need agents' coordination, and this prohibits the usage of single-agent planners that are unable to allow for coordination. To address these problems and enable emergent coordination in the decentralized MAPF, we propose to use the shared memory that can store coordination-related representations from agents' working memories. We combine these representations with path-planning information via the attention mechanism for effective task solving.

**Communication-based Coordination in MARL**    Inter-agent communication represents a natural solution to coordination challenges by allowing agents to share information about their experiences, observations, or plans. Communication-based approaches face fundamental trade-offs between coordination effectiveness and practical constraints. Methods like DCC (Ma et al., 2021b) enable selective information sharing but require domain-specific design of what information to communi-

cate. MAMBA (Egorov & Shpilman, 2022) introduces discrete communication protocols but suffers from limited message expressivity. SCRIMP (Wang et al., 2023) applies transformer architectures to scalable communication but requires significant bandwidth and fails under communication constraints.

**Memory Methods and Coordination**    Memory-related approaches in RL traditionally focus on enabling single agents to handle partial observability by maintaining historical information like RRNN (Santoro et al., 2018) and RATE (Cherepanov et al., 2024). Recent transformer-based approach ATM (Yang et al., 2022) extends these ideas to maintain recurrent memory states across long episodes. However, existing memory approaches treat each agent's memory as private, missing opportunities for coordination through shared representations. While ATM introduces transformer-based memory to MARL, it maintains separate memory buffers for each agent without enabling cross-agent information flow. Our approach differs fundamentally by treating memory as a coordination mechanism rather than just a solution to partial observability. By pooling individual memories into a shared workspace, we enable information exchange through learned representations rather than specifically crafted messaging protocols.

SRMT addresses the limitations of existing coordination approaches by combining the expressivity of learned representations with the scalability of decentralized execution. It also extends the memory transformers research (Burtsev et al., 2020; Bulatov et al., 2022; Sagirova & Burtsev, 2022; Yang et al., 2022; Sagirova & Burtsev, 2023) to multi-agent settings. Compared to the related communication works, SRMT requires no domain-specific message design. Unlike centralized approaches, SRMT maintains scalability by avoiding global controllers. Unlike existing memory methods, SRMT enables coordination through shared memory workspace rather than treating memory as purely private information storage.

## 3 PRELIMINARIES

**Multi-agent Pathfinding**    We consider two MAPF settings: classical and lifelong. In classical MAPF (Stern et al., 2019), $N$ agents interact in the two-dimensional environment represented as a graph $G = (V, E)$ with the vertices corresponding to the locations and the edges to the transitions between these locations. The timeline consists of discrete time steps. The agents' interaction episode ends when the predefined time step, called episode length, is reached. The episode can also end before this time step if all agents reach their goal locations. At the beginning of the episode, each agent $i$ is given a start location $s_i \in V$ and a goal location $g_i \in V$ to be reached until the end of the episode. At each time step, an agent performs an action to stay in its current location or move to an adjacent one. An agent's plan is a sequence of actions transferring the agent between the start and target vertices. Two distinct plans can have one of the two types of conflicts: a vertex conflict, where the agents occupy the same vertex at the same time step, and an edge conflict, where the agents use the same edge at the same time step. The MAPF task is to find a set of $N$ conflict-free plans. Typically, the task objective is to minimize the *Sum-of-Costs (SoC)* – the total cost of all agents' plans, where plan's cost is defined as the number of actions comprising it. In practice, if, according to the predicted movements, agents may collide, the final decision will be to retain their current positions until the next step. Lifelong MAPF (LMAPF) is an extension of classical setting where if an agent reaches its goal before the episode ends, it is immediately assigned to another one and continues its operation. We assume the goals are assigned via an external procedure, and the agents' behavior does not affect the goal assignments.

**MAPF as a sequential decision-making problem**    Though MAPF is usually treated as a planning problem, it can also be framed as a sequential decision-making (SDM) problem. The SDM task is to construct an *individual* policy $\pi$ for each agent that takes a graph and a history of observations as input and outputs a distribution over actions. At each time step, an agent samples an action from the policy $\pi$ distribution and executes it in the environment. This continues until the end of the episode. To ensure scalability, we consider decentralized policy where each agent chooses its action independently of the other agents. We also consider our decentralized agents to be Partially Observable (PO): each agent has complete knowledge of the graph $G$, but it can observe the other agents only locally in the area of the size $m \times m$, centered at the agent's current position. The

agent observes the locations of the other agents and the anonymized targets and does not know their intended paths.

**Decentralized Partially Observable MDP** We model MAPF task as a Partially Observable multi-agent Markov Decision Process (Dec-POMDP) (Oliehoek et al., 2016) presented by a tuple $M = \langle S, U, A, P, R, O, \mathcal{O}, \gamma \rangle$, where $s \in S$ are the environment states, $U = \{1, \ldots, n\}$ represents the agents, $A$ is the set of possible actions with action of agent $u$ denoted as $a^{(u)}$. At each time step, agents form a joint action $a^{(U)} \in A^n$ and the environment state is updated with transition function

$$P(s' \mid s, a^{(U)}) : S \times A^n \times S \to [0, 1]. \tag{1}$$

After all agents have performed their actions, each of them receives the following scalar reward

$$R(s, u, a^{(U)}) : S \times U \times A^n \to \mathbb{R}, \tag{2}$$

reflecting the agent's performance during the previous step. The future rewards are discounted by a factor $0 \leq \gamma \leq 1$, defining their importance. Before the next step, each agent also receives its *local* observation $o^{(u)} \in O$ based on the following global observation function

$$\mathcal{O}(s, a) : S \times A \to O. \tag{3}$$

To estimate a learnable stochastic policy $\pi^{(u)}$, each agent preserves the historical action-observation sequence $h^{(u)} \in H = (O \times A)$

$$\pi^{(u)}(a^{(u)} \mid h^{(u)}) : H \times A \to [0, 1]. \tag{4}$$

The training objective is to maximize the expected cumulative reward over time. In this work, we follow a common assumption in MAPF (Skrynnik et al., 2024b;c; Egorov & Shpilman, 2022) to consider homogeneous agents. To notice, homogeneity is the most theoretically demanding scenario for testing coordination in MAPF, because it represents a highly challenging problem – when all agents have identical capabilities and observations, they cannot rely on role differentiation or complementary skills or features to achieve coordination. They must learn to coordinate purely through emergent communication and implicit role assignment.

To compare different policies and measure the episode's success, we use the following metrics. For classical MAPF, we calculate *Cooperative Success Rate (CSR)* – a binary measure indicating whether all agents reached their goals before the episode ended, and *Individual Success Rate (ISR)* – the fraction of the agents that achieved their goals during the episode. For LMAPF, we compute *Throughput* — the ratio of the number of goals reached by all agents to episode length.

**Proximal Policy Optimization** PPO (Schulman et al., 2017) is a class of policy-gradient methods that constrains the policy change in a small range using a clip. It optimizes the following clipped surrogate objective:

$$\mathcal{L}(\theta) = \hat{\mathbb{E}}_t \left[ \min \left( \frac{\pi_\theta(a_t|s_t)}{\pi_{\theta_{old}}(a_t|s_t)} \hat{A}(s_t, a_t), \text{clip} \left( \frac{\pi_\theta(a_t|s_t)}{\pi_{\theta_{old}}(a_t|s_t)}, 1 - \epsilon, 1 + \epsilon \right) \hat{A}(s_t, u_t) \right) \right], \tag{5}$$

where $\theta_{old}$ are the policy parameters before the update, $\hat{A}(s_t, u_t)$ is an approximation of advantage function, $\epsilon$ is the clipping hyperparameter.

## 4 SHARED RECURRENT MEMORY TRANSFORMER

The Shared Recurrent Memory Transformer (SRMT) is a memory-augmented transformer that allows $N$ decentralized agents to broadcast their recurrent memories to a global workspace. Agents query the workspace via cross-attention for coordination (Fig. 1). For each agent $i \in [1, \ldots, N]$ at time step $t \in [1, \ldots, T]$, the SRMT takes as input a sequence $X_{i,t} = [\text{mem}_{i,t}, H_{i,t}, \text{obs}_{i,t}]$, where $\text{mem}_{i,t}$ is the agent's current memory state, $H_{i,t} = [\text{obs}_{i,t-h}, \ldots, \text{obs}_{i,t-1}]$ is a history of $h$ previous encoded observations, and $\text{obs}_{i,t}$ is the encoded representation of the current observation. We overload the $\text{obs}_{i,t}$ to denote both the raw observation and its encoding produced by the Spatial Encoder, for brevity. Firstly, the agent processes its individual inputs with multi-head self-attention:

$$A_{i,t}^{\text{self}} = X_{i,t} + \text{Self-Attention}(\text{LN}(X_{i,t})). \tag{6}$$

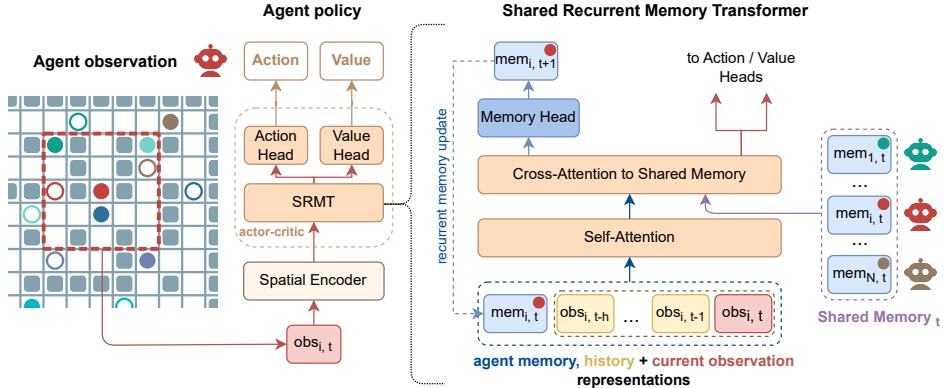

Figure 1: **Shared Recurrent Memory Transformer.** At each step $t$, SRMT pools agents' recurrent memories $\text{mem}_{i,t}$ into a shared memory and provides the *global access* to this space via the cross-attention. *Agent observation*: a local grid containing static obstacles (gray squares), other agents (solid circles), and anonymized goals (empty circles). *Agent policy*: SRMT is integrated into the actor-critic pipeline; actor and critic share weights. The observation $\text{obs}_{i,t}$ goes through the encoder, SRMT, and dedicated heads to generate action and value predictions. *SRMT architecture*: a self-attention layer combines the agent's actual memory state with its historical and current observations. The cross-attention layer allows agents to query the shared memory state. Resulting representations are used for recurrent memory update and action-value prediction for policy training.

Secondly, the global context is incorporated into the agent's decision-making process via multi-head cross-attention between $A_{i,t}^{\text{self}}$ and the shared memory state $\text{SM}_t = [\text{mem}_{1,t}, \text{mem}_{2,t}, \ldots, \text{mem}_{N,t}]$:

$$A_{i,t}^{\text{cross}} = \text{LN}(A_{i,t}^{\text{self}} + \text{Cross-Attention}(\text{LN}(A_{i,t}^{\text{self}}), \text{SM}_t)). \qquad (7)$$

Then, a position-wise feed-forward layer with residual and LayerNorm gives:

$$\text{out}_{i,t} = \text{LN}(A_{i,t}^{\text{cross}} + \text{FFN}(A_{i,t}^{\text{cross}})). \qquad (8)$$

The last element of $\text{out}_{i,t}$ is used to predict action and value for agent training:

$$\text{action}_{i,t} = \text{ActionHead}(\text{out}_{i,t}[-1]), \text{value}_{i,t} = \text{ValueHead}(\text{out}_{i,t}[-1]). \qquad (9)$$

To prepare for the next time step, the observation history window rolls forward: $H_{i,t+1} = [\text{obs}_{i,t-h+1}, \ldots, \text{obs}_{i,t-1}, \text{obs}_{i,t}]$, and the agent's memory stored in first element of $\text{out}_{i,t}$, is passed through the memory head dense layer to generate the updated memory state:

$$\text{mem}_{i,t+1} = \text{MemoryHead}(\text{out}_{i,t}[0]). \qquad (10)$$

After all agents have produced their predictions for step $t$, the shared memory is updated:

$$\text{SM}_{t+1} = [\text{mem}_{1,t+1}, \text{mem}_{2,t+1}, \ldots, \text{mem}_{N,t+1}]. \qquad (11)$$

The policy network comprises (i) a spatial encoder of ResNet (He et al., 2016) blocks followed by an MLP, (ii) the SRMT core working as actor-critic with shared weights, and (iii) separate dense layers for action and value heads. The encoder receives a local observation tensor of shape $3 \times m \times m$, where $m$ is the observation range. The observation encodes obstacles and current path, other agents, and their anonymized targets. The current path is the shortest path to the goal at each time step. The network weights are shared between agents to accelerate training. Details regarding model architecture, sizes, hyperparameters, learning curves, and training are listed in Appendix D.

## 5 EXPERIMENTS AND RESULTS

**Baselines** In our experiments, we use two cooperative baselines (MAMBA, QPLEX) to compare CTDE to our decentralized method with global shared space for communication, three memory-augmented architectures (ATM, RATE, RRNN) to assess the effectiveness of the SRMT memory mechanism, and three advanced path planners for LMAPF evaluations (centralized state-of-the-art RHCR and decentralized search-and-learning MATS-LP and Follower). The summary of our baselines is presented in Table 1. We provide the discussion of each method in detail in Appendix G. As an additional baseline, we consider RMT – the version of SRMT that uses agents' individual memories without sharing them.

(a) Bottleneck  (b) Maze  (c) MovingAI  (d) Puzzle  (e) Random  (f) Warehouse

Figure 2: **Environments for our experiments.** (a) Bottleneck: two agents start in rooms opposite their goals and should coordinate passing through the corridor. (b)-(f) Maps from POGEMA benchmark. Agents are solid-colored circles, and their goals are correspondingly colored empty circles.

**Classical MAPF on Bottlenecks**  For experiments, we utilize the POGEMA (Skrynnik et al., 2024a) framework, where a two-dimensional environment is represented as a grid composed of obstacles and free cells (Fig. 2). Using the framework, we create a two-agent Bottleneck task (Fig. 2a) as a minimal proxy for more complex tasks. It isolates the core coordination challenge: two agents starting in opposite rooms must pass through a narrow corridor to reach their goals in the opposite rooms, avoiding blocking one another. This scenario requires implicit negotiation over passage priority. When the corridor length exceeds agents' observation range, the task becomes partially observable, forcing agents to rely on learned coordination strategies rather than reactive responses to immediate observations. Also, lengthening the corridor allows scalability assessment.

Table 1: **Baselines summary.** [*]The original method was tested in single-agent environments. For fair comparison in a multi-agent setting, we trained and executed it in a decentralized way.

| Method | Group | Paradigm |
|---|---|---|
| MAMBA (Egorov & Shpilman, 2022) | cooperation | CTDE |
| QPLEX (Wang et al., 2021) | cooperation | CTDE |
| Follower (Skrynnik et al., 2024b) | adv. path plan. | decentr. |
| MATS-LP (Skrynnik et al., 2024c) | adv. path plan. | decentr. |
| RHCR (Li et al., 2021b) | adv. path plan. | centr. |
| ATM (Yang et al., 2022) | memory | decentr. |
| RATE (Cherepanov et al., 2024) | memory | decentr.[*] |
| RRNN (Santoro et al., 2018) | memory | decentr.[*] |
| **SRMT (ours)** | mem. for coop. | decentr. |

We test three reward functions that probe different aspects of coordination learning. *Directional* reward function provides guidance towards the goal, testing whether agents can coordinate while following environmental signals. *Moving Negative* reward function penalizes movement without the target location guidance, requiring agents to balance progress with coordination efficiency. *Sparse* reward function only gives a reward for goal achievement, testing whether agents can learn coordination purely from task completion without intermediate guidance (for detailed definitions see Table 8 in Appendix E).

Results are presented in Table 2. Communicative MAMBA and cooperative QPLEX completely fail across all rewards (0% success rates), indicating their state information-sharing methods cannot handle the implicit negotiation required in the task. Memory-based methods show mixed results: the private memory methods without information sharing (ATM, RRNN) cannot solve the coordination task, while RATE's decision transformer architecture lacks the real-time adaptation needed for dynamic coordination in the Moving Negative reward

Table 2: **SRMT effectively solves the Bottleneck task with different reward functions.** With Directional (positive when moved towards a goal and achieved it) reward, SRMT outperforms cooperative methods, ATM, and RRNN. For Moving Negative reward (movement penalty and no directional reward), SRMT and RMT outperform the other baselines. With a Sparse (on-goal only) reward, SRMT shows the best score. The values are averaged over multiple runs with different random seeds.

| Method | Directional | | Moving Neg. | | Sparse | |
|---|---|---|---|---|---|---|
| | CSR ↑ | ISR ↑ | CSR ↑ | ISR ↑ | CSR ↑ | ISR ↑ |
| SRMT | **1.0**±0.0 | **1.0**±0.0 | **0.8**±0.4 | **0.9**±0.2 | **1.0**±0.0 | **1.0**±0.0 |
| MAMBA | 0.0±0.0 | 0.0±0.0 | 0.0±0.0 | 0.0±0.0 | 0.0±0.0 | 0.0±0.0 |
| QPLEX | 0.0±0.0 | 0.1±0.0 | 0.0±0.0 | 0.3±0.0 | 0.0±0.0 | 0.0±0.0 |
| RMT | **1.0**±0.0 | **1.0**±0.0 | **0.8**±0.4 | **0.9**±0.2 | 0.9±0.3 | 0.9±0.2 |
| ATM | 0.0±0.0 | 0.0±0.0 | 0.1±0.0 | 0.3±0.0 | 0.0±0.0 | 0.1±0.0 |
| RATE | **1.0**±0.0 | **1.0**±0.0 | 0.6±0.5 | 0.7±0.5 | 0.9±0.3 | 0.9±0.3 |
| RRNN | 0.0±0.0 | 0.1±0.1 | 0.0±0.0 | 0.0±0.0 | 0.0±0.0 | 0.0±0.0 |

case. SRMT achieves 100% scores under Directional and Sparse rewards. The comparison with RMT reveals that both methods achieve similar success under path-guiding rewards, but SRMT outperforms RMT under Sparse reward (100% vs 90% CSR/ISR), demonstrating that shared

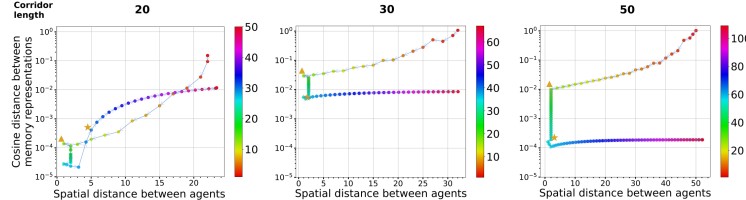

Figure 3: **SRMT memory distances are aligned with inter-agent distances and coordination.** The figure shows how cosine distances between agents' memories are related to the Euclidean distances between agents for the Bottleneck task. As agents approach each other, memory distances decrease, aligning memory representations for coordination. After meeting (triangle mark), while agents pass corridor, their representations are steadily close, suggesting the strategies are synced. After the corridor passage, coordination is not needed, so the memory distances increase. Star marks first goal achievement step. Color bar shows the step number.

memory enables coordination with minimal environmental feedback. SoC scores provided in Table 7 Appendix E support SRMT's superior performance.

**Coordination Mechanism Analysis** Figure 3 shows the correlations between memory cosine distances and agent spatial distances for Bottlenecks with different corridor lengths, revealing how SRMT learns to encode coordination intentions. At the start, agents move to each other, and memory distances decrease, indicating that agents learn to align their internal representations when coordination is required. This synchronization occurs before spatial conflict, showing predictive coordination rather than reactive responses. After meeting, agents' memory representations remain synchronized while agents coordinate movement in the corridor. The stable low memory distance during this critical phase can be considered as agents maintain a shared understanding of their strategies. After the coordinated corridor passage, memory distances increase as agents pursue independent goals, indicating that coordination is not demanded to solve this part of the task. The detailed heatmaps of cosine distances between agent memory states during the episode are presented in Appendix H.

**Bottlenecks Generalization** Figure 4 shows that SRMT coordination strategies learned on corridor lengths 3-30 (used for training) generalize to lengths up to 1000 cells. To ensure agents can reach their goals, the episode length is set to $2 \cdot \text{corridor\_len} + 100$. This evaluation demonstrates that SRMT agents transfer coordination across spatial scales and learn general principles of implicit negotiation rather than memorizing specific spatial patterns. Under Sparse rewards, SRMT maintains perfect scores up to 400-cell corridors with further slight degradation (100% to 80% CSR). For the Moving Negative reward function, SRMT shows the top-1 scores for all corridor lengths up to 1000. While RMT maintains the scores under Sparse reward, its performance degrades more severely in Moving Negative reward case, confirming that shared memory is critical when the combination of immediate path-planning

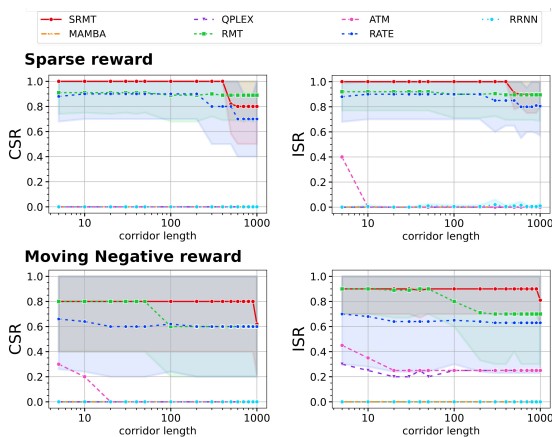

Figure 4: **SRMT generalizes on corridors up to 1000 cells.** In Moving Negative case, requiring immediate path adaptation in addition to coordination, SRMT performs the best.. In Sparse reward case, SRMT leads up to 400 with a further slight drop. All methods were trained on 3-30 cell corridors and evaluated on passages up to 1000. Shaded area indicates 95% confidence intervals.

feedback from reward and long-time coordination is needed for task solving. The evaluation of other reward functions and wide error bar range explanation can be found in Appendix E.

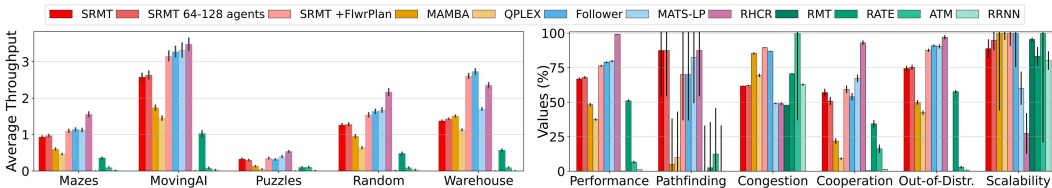

Figure 5: **SRMT outperforms cooperative and memory baselines on POGEMA maps.** *Left:* SRMT robustly generalizes over various types of maps, outperforming cooperative QPLEX and memory models. On the Warehouse map, SRMT with heuristic path planning exceeds baselines, including the centralized non-trainable planner RHCR, except Follower. *Right*: SRMT and its variants demonstrate competitive performance in POGEMA metrics, particularly in Scalability and Pathfinding. Error bars indicate 95% confidence intervals.

**Ablation Study** On the Bottleneck task, we also conduct an ablation study. We test the SRMT memory update rule by comparing it with the ones used in RMT and RATE (using individual memories only). We remove the observation history (No History) and the Memory Head layer (No MemHead) from SRMT to prove their impact on performance. We confirm the importance of memory augmentation by comparing SRMT with a memory-less transformer (Attention), the direct connection between the spatial encoder

Table 3: **SRMT ablation study resuts.**

| Method | Directional | | Moving Neg. | | Sparse | |
|---|---|---|---|---|---|---|
| | CSR ↑ | ISR ↑ | CSR ↑ | ISR ↑ | CSR ↑ | ISR ↑ |
| SRMT | **1.0**±0.0 | **1.0**±0.0 | **0.8**±0.4 | **0.9**±0.2 | **1.0**±0.0 | **1.0**±0.0 |
| No SM: RMT | **1.0**±0.0 | **1.0**±0.0 | **0.8**±0.4 | **0.9**±0.2 | 0.9±0.3 | 0.9±0.2 |
| No SM: RATE | **1.0**±0.0 | **1.0**±0.0 | 0.6±0.5 | 0.7±0.5 | 0.9±0.3 | 0.9±0.3 |
| No History | **1.0**±0.0 | **1.0**±0.0 | 0.1±0.2 | 0.3±0.3 | 0.6±0.5 | 0.8±0.3 |
| No MemHead | 0.0±0.0 | 0.0±0.0 | 0.2±0.4 | 0.2±0.4 | 0.0±0.0 | 0.0±0.0 |
| Attention | **1.0**±0.0 | **1.0**±0.0 | 0.4±0.5 | 0.6±0.4 | 0.7±0.5 | 0.8±0.4 |
| Empty | 0.8±0.5 | 0.9±0.2 | 0.0±0.0 | 0.3±0.3 | 0.7±0.5 | 0.7±0.4 |
| RNN | **1.0**±0.0 | **1.0**±0.0 | 0.4±0.5 | 0.4±0.5 | 0.7±0.5 | 0.8±0.2 |

and action and value heads (Empty), and a replacement of SRMT with a GRU unit (Cho et al., 2014) (RNN) in the policy network. Table 3 shows significant performance drops, especially in the Sparse reward, demonstrating the effectiveness and efficiency of the proposed architecture. In Appendix F, we also added observation history and memory length ablations.

**Lifelong MAPF** LMAPF presents significantly more complex coordination challenges than the Bottleneck task: agents are required (i) to adapt their strategies and re-plan paths as goals change, requiring flexible negotiation patterns; (ii) effectively coordinate to avoid the high number of conflicts in densely populated, structured environments; (iii) maintain stable coordination strategies over extended periods while adapting to changing conditions; (iv) remain fully decentralized and rely on local interactions only. We use cooperative, advanced path planning, and memory baselines (Tab. 1). We train SRMT and baselines on maze environments (Fig. 2b) with the following reward function: agent receives a reward $r = 0.01$. for following the planned path and $r = 0$ otherwise. We also consider an advanced Heuristic Path Decider planning from the Follower that combines the shortest path with heuristic search to find evenly dispersed paths. We compare SRMT trained with 64 agents (SRMT), SRMT with a mixture of 64 and 128 agents (SRMT 64-128), and SRMT with Heuristic Path Decider (SRMT+FlwrPlan). The advanced path-planning baselines (Follower, MATS-LP, RHCR) are considered as the performance upper bound, as they are focused on optimal path prediction rather than agents' cooperation and communication. For example, RHCR is a centralized oracle that has full environment information and replans at every step, marking an unreachable theoretical upper bound for decentralized RL methods. Figure 5 demonstrates LMAPF evaluation results across diverse environments using the Average Throughput metric of LMAPF performance (left) and specific aggregated metrics from POGEMA benchmark (right). We assess the results in three directions: generalization on unknown maps, coordination, and scalability in terms of environment and agent population size. We discuss each direction separately below.

*Generalization* is analyzed using the *Average Throughput* metric on each type of maps, *Performance* (average throughput normalized by the best value among methods on Random ($20 \times 20$) and Mazes ($21 \times 21$) maps), *Out-of-Distribution* (Performance score on unseen $64 \times 64$ MovingAI-tiles), *Pathfinding* (binary value of optimality of a single agent path on large $256 \times 256$ MovingAI (Stern et al., 2019) maps). SRMT robustly generalizes to different unseen types of maps, indicating that learned coordination principles transfer across different obstacle configurations. SRMT outperforms

cooperative and memory methods, except MAMBA on the Warehouse map ($1.38 \pm 0.02$ Average Throughput for SRMT, $1.43 \pm 0.02$ for SRMT 64-128, and $1.50 \pm 0.03$ for MAMBA). The Warehouse evaluation uses a single obstacle configuration in contrast to the rest of the evaluation tasks that use 8-128 different obstacle configurations. Such a minimalistic setting leads to a significantly reduced variety in evaluation data, contributing to the difference in SRMT and MAMBA scores.

For *Coordination* assessment, we analyzed *Cooperation* (average throughput on the $5 \times 5$ Puzzle maps crafted with cooperation-requiring patterns) and *Congestion* (the ratio of the average local agent density in each observation to the global density on Warehouse maps) metrics. On cooperation-demanding Puzzle maps, SRMT outperforms cooperative, memory baselines, and Follower, suggesting that learned coordination and communication through the shared memory space provides advantages over centralized training, individual memory, and planning heuristics approaches. Regarding Congestion, SRMT with planning heuristics (SRMT+FlwrPlan) improves relative agents' density, leading to the top-2 score. Combining specialized planning with shared recurrent memory also helps SRMT to surpass baselines, including the centralized oracle RHCR, by average throughput (see Warehouse scores in Fig. 5, left). This indicates the complementary benefits of learning-based approaches and planning algorithms.

*Scalability* is important to prove the practical efficiency of the method. We analyze it with *Scalability* score (the ratio between relative runtimes and relative numbers of agents on the Warehouse map for growing population size). We additionally provide the average throughput scores on MovingAI maps in Fig. 12 Appendix E to demonstrate that SRMT successfully scales with an increasing number of agents on large-sized maps. While SRMT requires additional time and computation to update and broadcast agents' memories compared to no-memory no-communication methods (QPLEX, Follower), it outperforms individual memory-based RRNN and RATE, demonstrating that shared memory provides better coordination-to-computation trade-offs. Advanced planning heuristics and training with a mixed number of agents improve SRMT via selection optimization and training on larger agent populations, which predictably leads to better scalability: SRMT+FlwrPlan reaches 100%. To analyze the scalability with the environment size, we compare average throughput on the training Maze maps of size $65 \times 65$ to $256 \times 256$ MovingAI maps: SRMT outperforms cooperative and memory baselines, proving the effective scalability to larger unseen maps.

**Train and Inference Costs**  Table 4 provides the inference time and memory usage for the model with and without shared memory. SRMT was run in a lifelong regime on a single NVIDIA A100-40GB, 96 CPUs AMD EPYC 7352 24-Core Processor, single-environment single-thread setup. Shared memory addition has no significant effect on inference time, scaling almost linearly with the number of agents. However, a larger memory amount is required for 512 agents and more. We also compared SRMT training costs versus a transformer-based MAMBA on the

Table 4: **SRMT inference efficiency.**

| | With Shared Mem. | | No Shared Mem. | |
|---|---|---|---|---|
| Agents | Steps/sec | Mem., MB | Steps/sec | Mem., MB |
| 32 | 6.8 | 1370 | 7.7 | 1370 |
| 64 | 3.5 | 1370 | 3.9 | 1370 |
| 128 | 1.7 | 1370 | 2.0 | 1370 |
| 256 | 1.0 | 1754 | 1.0 | 1370 |
| 512 | 0.4 | 2906 | 0.5 | 1370 |
| 1024 | 0.1 | 7520 | 0.2 | 1376 |

Bottleneck task using a single Tesla P100, 80 Intel Xeon E5-2698 CPUs. SRMT training for 20M steps took 99 minutes, MAMBA training for 200k steps – 325 min. One training step for SRMT takes $5 \pm 2$ seconds and for MAMBA $13 \pm 1$ seconds. Maximal memory allocated for training SRMT – 4131 MB, for MAMBA – 2801 MB.

## 6   CONCLUSION

In this paper, we introduced a novel Shared Recurrent Memory Transformer (SRMT) architecture to enhance coordination in multi-agent systems. SRMT enables inter-agent information exchange via shared memory and action coordination through unconstrained communication. The experimental results on the Bottleneck task show that SRMT consistently outperforms baselines and effectively coordinates even in the sparse reward case. SRMT demonstrates scalability and robustness, allowing agents to generalize to maps with significantly longer corridors than those seen during training. On POGEMA maps, SRMT is competitive with various recent MARL, hybrid, and planning-based methods. Our findings highlight the potential of SRMT for coordination in multi-agent pathfinding and multi-agent reinforcement learning.

## REPRODUCIBILITY STATEMENT

All reported metrics are averaged across multiple runs with different random seeds and reported with 95% confidence intervals. All hyperparameters are specified in Table 5. We describe training details and hardware used for experiments in Section D. We also provide the source code as a Supplementary Material with this submission to ensure reproducibility of results.

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

## A    THE USE OF LARGE LANGUAGE MODELS (LLMS)

We used LLMs only for text polishing and editing.

## B    LIMITATIONS

We evaluated SRMT using the POGEMA suite and a PPO actor-critic pipeline in a grid-based MAPF setting. This setup emphasizes SRMT's target capabilities, such as memory-based communication and coordination, and provides standardized, large-scale benchmarks. Although SRMT itself is task-agnostic, demonstrating its potential in other domains spaces, with heterogeneous agents, or under different training methods is a possible next step that can be explored further.

As in the majority of research related to Multi-Agent Pathfinding, in this work, we assume that the agents have flawless localization and mapping abilities. Our primary focus is on the decision-making aspect of the problem. We also consider that the agents execute actions accurately and that their moves are synchronized. Additionally, we treat obstacles as fixed elements of the environment. Finally, it is important to note that our approach, like other prominent learnable methods designed for (PO)-MAPF – such as PRIMAL (Sartoretti et al., 2019), PRIMAL2 (Damani et al., 2021), DHC (Ma et al., 2021a), and PICO (Li et al., 2022) – does not offer theoretical guarantees that agents will reach their destinations. However, extensive experimental evidence from our work and the referenced studies, demonstrates that these learnable methods are practically powerful and scalable solutions for complex MAPF problems.

## C    SRMT INNOVATIONS AND TECHNICAL CHALLENGES

SRMT, unlike a standard Transformer (Vaswani et al., 2017) applied to multi-agent settings, introduces the following key innovations:

- Shared memory pool architecture: unlike conventional transformers case, where each agent uses transformer independently, SRMT maintains a distributed but globally accessible memory pool. Each agent writes to its dedicated memory slot while being able to read from all other agents' slots via the cross-attention. This creates an implicit coordination channel without centralized decision-making.
- Each agent's SRMT module attends to other agents' memory states containing information from previous timesteps. This ensures decentralized execution—agents never access current-step information from others, only historical context. This is fundamentally different from CTDE, where a central critic conditions on joint actions.

One of the main technical challenges is the following: centralized methods fail to scale efficiently with the growing number of agents compared to decentralized approaches. However, decentralization in partially observable environments comes at a cost of no access to information about locations

Table 5: SRMT configuration and training hyperparameters.

| Parameter | MAPF | LMAPF |
|---|---|---|
| Optimizer | Adam | Adam |
| Learning rate | 0.00013 | 0.00022 |
| LR Scheduler | Adaptive KL | Constant |
| $\gamma$ (discount factor) | 0.9716 | 0.9756 |
| Recurrence rollout | 8 | - |
| Clip ratio | 0.2 | 0.2 |
| Batch size | 16384 | 16384 |
| Optimization epochs | 1 | 1 |
| Entropy coefficient | 0.0156 | 0.023 |
| Value loss coefficient | 0.5 | 0.5 |
| $GAE_{\lambda}$ | 0.95 | 0.95 |
| MLP hidden size | 16 | 512 |
| ResNet residual blocks | 1 | 8 |
| ResNet filters | 8 | 64 |
| Attention hidden size | 16 | 512 |
| Attention heads | 4 | 8 |
| GRU hidden size | 16 | - |
| Activation function | ReLU | ReLU |
| Network Initialization | orthogonal | orthogonal |
| Rollout workers | 4 | 8 |
| Envs per worker | 4 | 4 |
| Training steps | $2 \times 10^7$ | $10^9$ |
| Episode length | 512 | 512 |
| Observation patch | $5 \times 5$ | $11 \times 11$ |
| Number of agents | 2 | 64 |

and behavior of agents outside the tight observation window. This information is crucial for optimal solving of the path finding problem as agent have to avoid collisions and minimize congestion. SRMT presents a decentralized solution that efficiently scales with the size of the environment and the agent population size, and also SRMT provides agents with a coordination mechanism that allows implicit negotiation and coordination required for optimal path finding.

## D TRAINING DETAILS

The environments were created with POGEMA (Skrynnik et al., 2024a) framework. The Sample Factory codebase (Petrenko et al., 2020) was used for policy model training. The integration of SRMT into the PPO framework is presented in Algorithm 1. To implement the attention mechanisms in SRMT, we used the attention block from the Huggingface GPT-2[1]. Training hyperparameters for SRMT are listed in Table 5 and the number of trainable parameters for methods evaluated in the paper are presented in Table 6. A single Tesla P100 with 80 CPUs Intel(R) Xeon(R) CPU E5-2698 v4 @ 2.20GHz was used for training policy models on classical MAPF Bottlenecks for approximately 1 hour each, and a single NVIDIA A100-SXM4-40GB with 96 CPUs AMD EPYC 7352 24-Core Processor was used for the models' training and evaluations on LMAPF benchmark POGEMA for approximately 10 days each.

For training on Bottleneck task, we used 16 maps with corridor lengths drawn uniformly from 3 to 30 cells. The results for models trained with Sparse and Dense reward functions were averaged over 10 runs with different random seeds. The results of training policies with Directional and Directional Negative rewards were averaged over 5 runs with different random seeds, as they showed less variation during training.

---

[1] https://github.com/huggingface/transformers/blob/v4.43.3/src/transformers/models/gpt2/modeling_gpt2.py

---

**Algorithm 1** Pseudocode of integration SRMT into the PPO framework.

---

**for** iteration=1,2,... **do**
  ▷ *Run policy $\pi_{\theta_{old}}$ in environment for T time steps*    ◁
  **for** t=1,2,...,T **do**
    **for** i=1,...,n_agents **do**
      $o_{i,t} \leftarrow Spatial\_Encoder(obs_{i,t})$
      $seq_{i,t} \leftarrow [mem_{i,t}, o_{i,t-h}, \ldots, o_{i,t-1}, o_{i,t}],$    ▷ *h =length of historical obs. sequence*
      $out_{i,t}, mem_{i,t+1} \leftarrow SRMT(seq_{i,t}, shared\_mem_t)$
      $action_{i,t} \leftarrow Action\_Head(out_{i,t})$
      $value_{i,t} \leftarrow Value\_Head(out_{i,t})$
    $shared\_mem_{t+1} \leftarrow [mem_{1,t+1}, mem_{2,t+1}, \ldots, mem_{n\_agents,t+1}]$
  Compute advantage estimates $\hat{A}_1, \ldots, \hat{A}_T$
  Optimize surrogate L w.r.t. $\theta$
  $\theta_{old} \leftarrow \theta$

---

Table 6: Number of policy network parameters for models trained on Bottleneck task and POGEMA benchmark.

| Model | Bottleneck task | POGEMA benchmark |
|---|---|---|
| SRMT | 271k | 17M |
| RMT | 271k | - |
| Attention Core | 271k | - |
| Empty Core | 5k | - |
| RNN Core | 6k | - |
| MAMBA | 6M | 6M |
| QPLEX | 318k | 318k |
| ATM | 349k | 349k |
| RATE | 272k | 18M |
| RRNN | 8k | 7M |
| Follower | - | 5M |
| MATS-LP | - | 161k |
| RHCR | - | no training parameters |

For LMAPF, we trained SRMT and baselines on the set of 40 maze-like environments (Fig. 2b) of size $65 \times 65$ using the Follower (Skrynnik et al., 2024b) training scheme. We trained SRMT with 64 agents for 1B environment steps, SRMT with a mixture of 64 and 128 agents (SRMT 64-128) for 400M steps, and SRMT with Heuristic Path Decider (SRMT+FlwrPlan) with 64 agents for 600M environment steps. LMAPF results were averaged over three runs with different random seeds. Each run evaluation was first averaged over 10 different evaluation procedure random seeds. We carried out the grid search for the SRMT training entropy coefficient (range $[0.00001, 0.0003]$) and learning rate (range $[0.01, 0.05]$).

Figures 6 and 7 show the typical examples of learning progress of SRMT on Bottlenecks and POGEMA Mazes maps.

## E  EVALUATION SCORES

In this section, we provide additional experimental results for SRMT and baselines for Directional, Moving Negative, and Sparse rewards on Bottleneck task. For this task, we additionally measure a MAPF metric called *Sum-of-Costs (SoC)* – the total number of time steps taken by all agents to reach their goals. The resulting values of SoC are presented in Table 7 and Figure 11. Table 8 describes reward functions in detail. The results for Dense, Directional, and Directional Negative rewards on Bottleneck task are shown in Figures 8, 9, 10. SRMT has superior or competitive performance compared to the baselines.

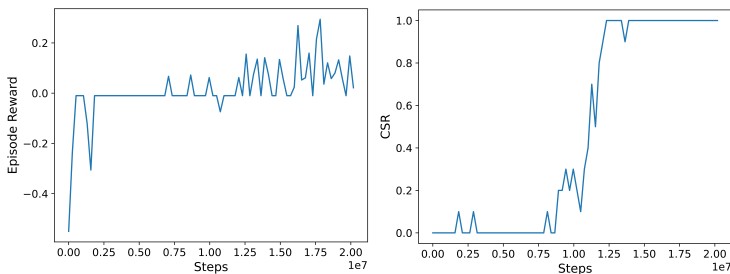

Figure 6: SRMT training progress on Bottleneck task.

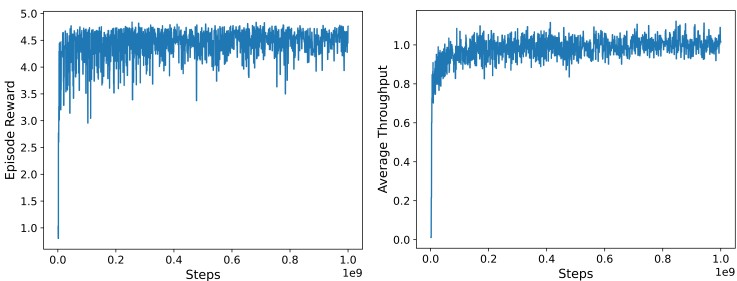

Figure 7: SRMT training progress on POGEMA Mazes task.

Table 7: Sum-of-Costs scores for SRMT and baselines for different reward functions. SRMT outperforms all the baselines. The values averaged over multiple runs with different random seeds.

| Method | Directional | Moving Negative | Sparse |
|---|---|---|---|
| SRMT | $\mathbf{126.7}_{\pm 0.6}$ | $\mathbf{167.2}_{\pm 77.7}$ | $\mathbf{130.9}_{\pm 5.7}$ |
| MAMBA | $306.0_{\pm 0.0}$ | $306.0_{\pm 0.0}$ | $306.0_{\pm 0.0}$ |
| QPLEX | $306.0_{\pm 0.0}$ | $306.0_{\pm 0.0}$ | $306.0_{\pm 0.0}$ |
| RMT | $127.0_{\pm 1.4}$ | $179.6_{\pm 77.0}$ | $145.3_{\pm 49.9}$ |
| ATM | $306.0_{\pm 0.0}$ | $293.3_{\pm 0.0}$ | $306.0_{\pm 0.0}$ |
| RATE | $126.8_{\pm 0.9}$ | $197.5_{\pm 95.2}$ | $151.4_{\pm 54.9}$ |
| RRNN | $306.0_{\pm 0.0}$ | $306.0_{\pm 0.0}$ | $306.0_{\pm 0.0}$ |

Table 8: Tested reward functions. We list the reward values for achieving the goal, moving on the path toward the goal, or taking other actions (moving not in the direction of the goal and staying in the same position).

| Type | On goal | Move towards goal | Else |
|---|---|---|---|
| Directional | +1 | +0.005 | 0 |
| Sparse | +1 | 0 | 0 |
| Dense | +1 | -0.01 | -0.01 |
| Directional Negative | +1 | -0.005 | -0.01 |
| Moving Negative | +1 | -0.01 | -0.01 for moving, -0.005 for holding |

**Reward design motivation**   The main reward functions used in our experiments (Directional, Moving Negative, Sparse), simulate the following realistic multi-robot deployment scenarios:

- Directional reward may be exemplified with automated baggage handling systems in the airport, where the only objective is a bag delivery (reward=+1) and moving toward gate (reward=+0.005) is slightly preferred to any other moving or staying on the same location (reward=0).

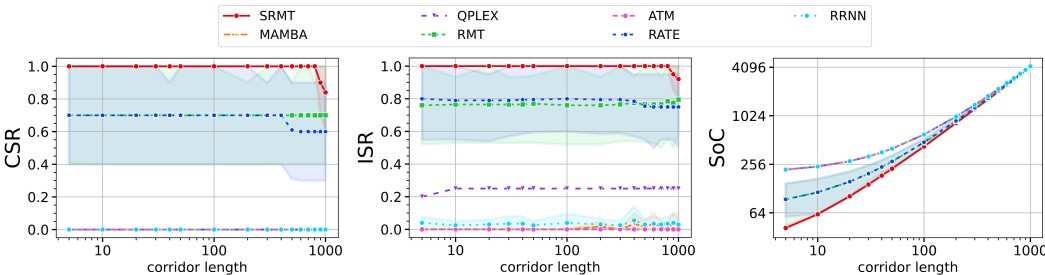

Figure 8: Trained with *Dense* reward, all models except empty core policy scale with enlarging corridor length. SRMT consistently outperforms baselines both in success rates and in the time needed to solve the task. The shaded area indicates 95% confidence intervals.

- Moving Negative reward models the warehouse automation scenario when thousands of autonomous robots with battery limitations are fetching items in fulfillment centers and their only goal is a task completion(reward=+1. When the agent moves its battery drains faster than when it is idle ((reward for moving=-0.01, for staying=-0.005).

- Sparse reward simulates the minimalistic setting for the warehouse robots where only task completion matters. This tests emergent coordination under sparse signals.

**Success rates error bar ranges** The reported success rates have a near-binary nature, leading to high variance values: agents either learn to cooperate and achieve scores close to the maximum or they fail, resulting in zero scores. Consequently, the methods that mostly succeed or fail have very small error values, but the methods that fail in some fraction of the training runs have wide error bar ranges. Figure 12 shows the evaluations of the methods from the POGEMA benchmark compared

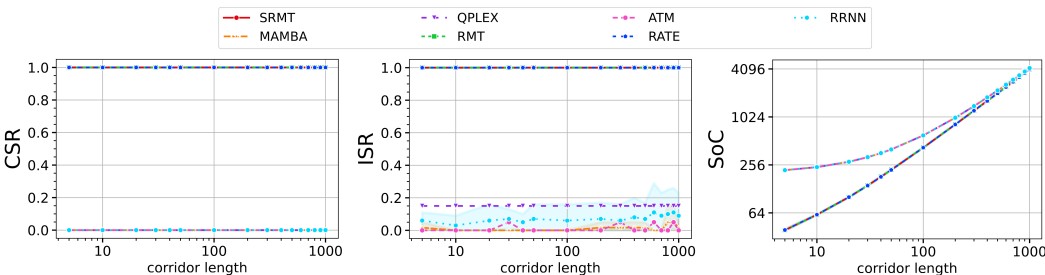

Figure 9: *Directional* reward training leads to all the methods preserving the scores for all tested corridor lengths. The shaded area indicates 95% confidence intervals.

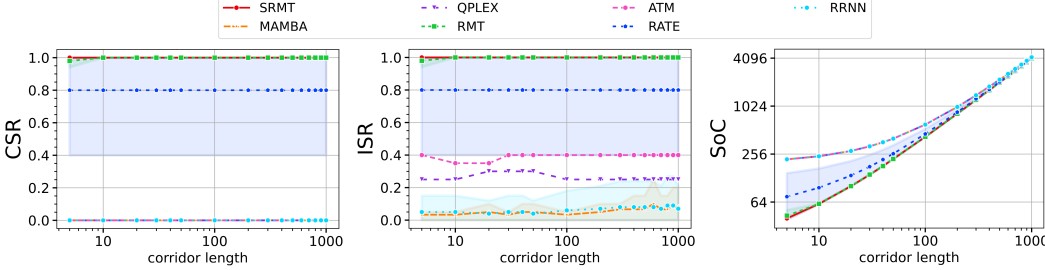

Figure 10: Results of training with *Directional Negative* reward. Vanilla attention fails to scale at corridor lengths of more than 400, compared to the SRMT which preserves the highest scores. That proves the sufficiency of the proposed SRMT architecture. The shaded area indicates 95% confidence intervals.

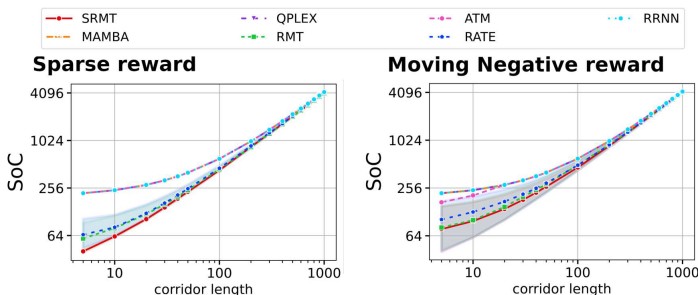

Figure 11: SoC scores for Sparse and Moving Negative reward functions on Bottlenecks. The shaded area indicates 95% confidence intervals.

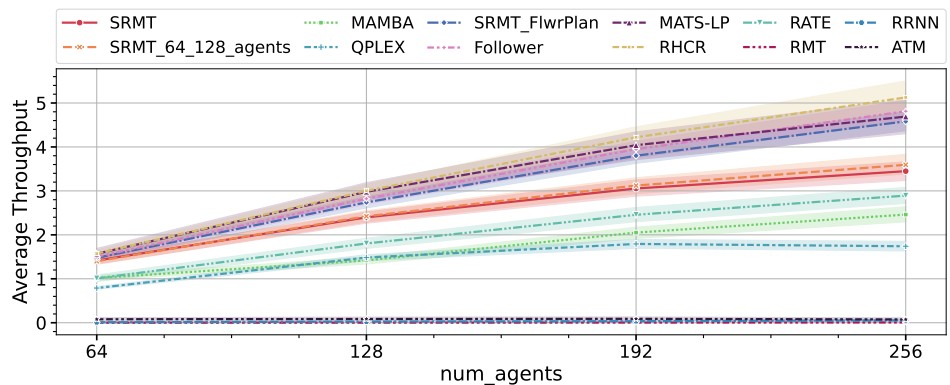

Figure 12: The evaluation of scalability of SRMT on MovingAI maps from POGEMA benchmark. The shaded area indicates 95% confidence intervals.

Table 9: The effect of observation history length on SRMT success scores for Bottleneck task with Moving Negative reward. The values are averaged over five runs with different random seeds. We provide the standard deviation values for each score and highlight the best values in bold.

| History length | CSR | ISR | SoC |
|---|---|---|---|
| 4 | $0.58_{\pm 0.43}$ | $0.64_{\pm 0.41}$ | $208.58_{\pm 78.84}$ |
| 8 | $\mathbf{0.80}_{\pm 0.40}$ | $\mathbf{0.90}_{\pm 0.20}$ | $\mathbf{167.23}_{\pm 69.51}$ |
| 16 | $0.56_{\pm 0.38}$ | $0.67_{\pm 0.25}$ | $235.16_{\pm 62.11}$ |
| 32 | $0.62_{\pm 0.41}$ | $0.69_{\pm 0.39}$ | $209.19_{\pm 69.21}$ |

to SRMT when evaluated on MovingAI maps with different numbers of agents equal to or greater than the ones used for training. SRMT was trained with 64 agents, SRMT_64_128 with a mixture of 64 and 128 agents. The results show that both SRMT models consistently outperform cooperative baselines (MAMBA and QPLEX). Also, all non-zero-scored methods except QPLEX successfully scale and improve the average throughput with the growing number of agents.

# F  ABLATION STUDY

To conduct the additional ablation experiments, we use the Bottleneck environment.

With SRMT we test different observation history and agent memory sequence lengths using the Moving Negative reward function (see Tables 9 and 10). The results show that smaller memory sizes improve the agent's performance. Memory size 4 shows slightly better scores than memory size 1, but within the standard deviation range. Also, memory size 1 helps to improve the attention mechanism computation efficiency by using the shorter input sequence.

Table 10: The SRMT agent memory sequence length ablation for Bottleneck task with Moving Negative reward. The values are averaged over five runs with different random seeds. We provide the standard deviation values for each score and highlight the best values in bold.

| Memory size | CSR | ISR | SoC |
|---|---|---|---|
| 1 | $0.80_{\pm 0.40}$ | $0.89_{\pm 0.20}$ | $167.23_{\pm 69.51}$ |
| 4 | $\mathbf{0.93}_{\pm 0.14}$ | $\mathbf{0.95}_{\pm 0.10}$ | $\mathbf{150.49}_{\pm 43.79}$ |
| 8 | $0.60_{\pm 0.49}$ | $0.68_{\pm 0.41}$ | $206.57_{\pm 81.67}$ |
| 16 | $0.44_{\pm 0.46}$ | $0.64_{\pm 0.31}$ | $231.85_{\pm 80.18}$ |

## G  BASELINE METHODS

The summary of methods used as baselines compared to SRMT is presented in Table 11.

## H  MEMORY ANALYSIS

To further analyze the relations between agents' memory representations, we provide the heatmaps of cosine distances between agent memory states during the episode. Figures 13 and 14 show how SRMT memory states are related to each other. Figures 15 and 16 depict the paired distances between memory vectors on different time steps in the episode for both agents.

Table 11: Baseline methods used for comparison with SRMT. The Group column shows the primary focus of the method: cooperation, planning, memory. The Paradigm denotes if the centralized training with decentralized execution (CTDE), or a decentralized or centralized paradigm was used for training and execution. *The original method was tested in single-agent environments, but for fair comparison with other multi-agent methods, we trained and executed it in a decentralized way.

| Method | Group | Paradigm | Description |
|---|---|---|---|
| MAMBA | cooperation | CTDE | Employs a transformer-based communication module to update agents' world models to achieve better cooperation. |
| QPLEX | cooperation | CTDE | During centralized training, the model learns to factorize a joint action-value function. In the decentralized execution, duplex dueling connects local and joint action-value functions, allowing agents to act independently. |
| Follower | advanced path planning | decentralized | Uses a centralized path planning with heuristic search tailored to avoid agents' collisions and trains a policy neural network without external guidance. |
| MATS-LP | advanced path planning | decentralized | Employs a learnable policy with a modification of neural Monte Carlo Tree Search for the improved path planning. |
| RHCR | advanced path planning | centralized | A search-based centralized path planner that does not require training. The paths are re-planned according to the predefined schedule, resolving collisions within the current planning window only. |
| ATM | memory | decentralized | Each agent maintains a historical sequence of memory states and updates them sequentially by a transformer network. |
| RATE | memory | decentralized* | The memory-dedicated module is added to a Decision transformer to store the historical information about long segmented trajectories. |
| RRNN | memory | decentralized* | The attention mechanism is used to update the memory state given input data. The model outputs are predicted by the modified LSTM that uses memory matrix values as recurrent cell states. |
| SRMT | memory for cooperation | decentralized | To predict next action, each agent uses a personal recurrently updated memory state and communicates with others through a shared memory. Agents learn read-write operations with memory in a shared space to retain information during episode and enable effective individual and collective decision-making. |

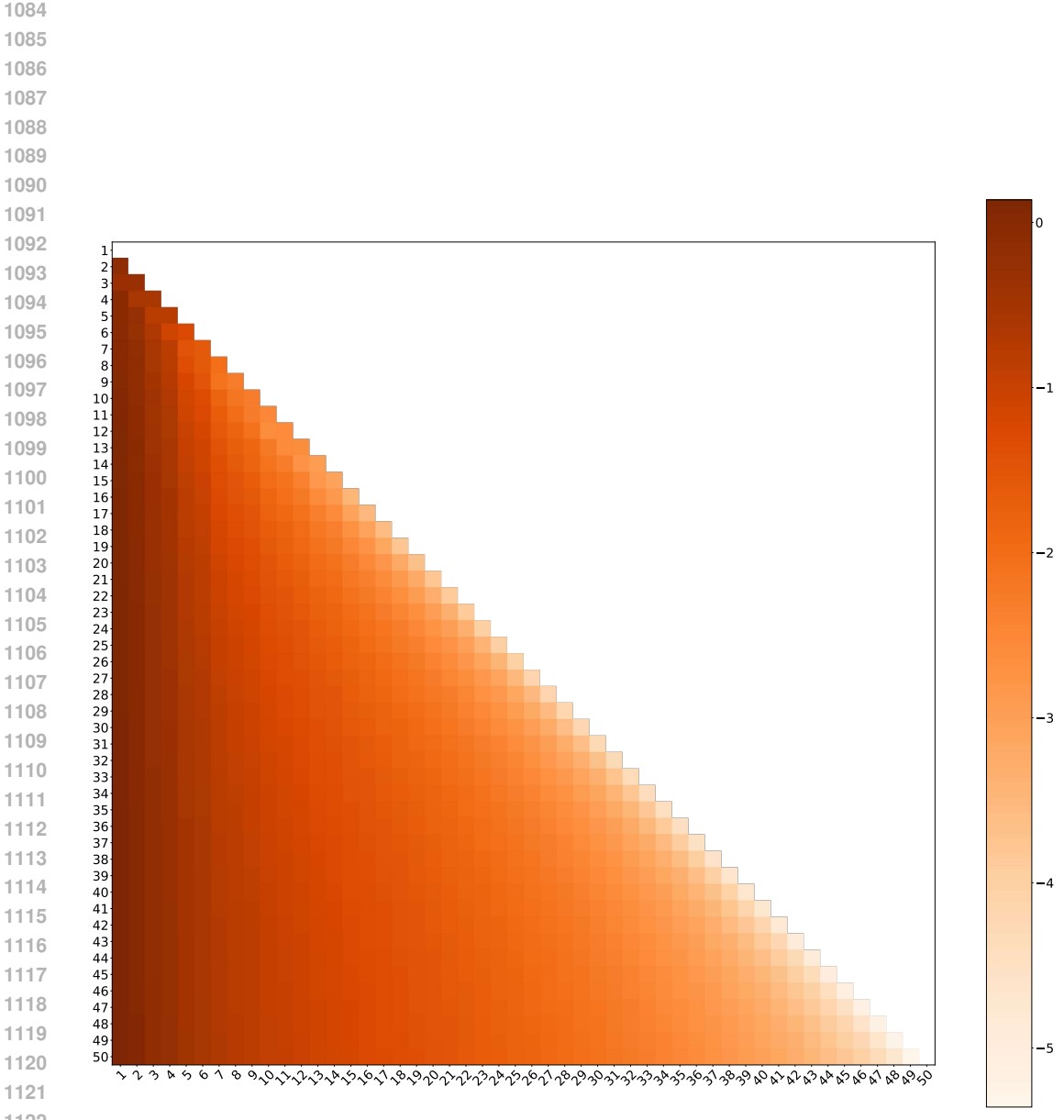

Figure 13: SRMT agent 1 heatmap of distances between memory states.

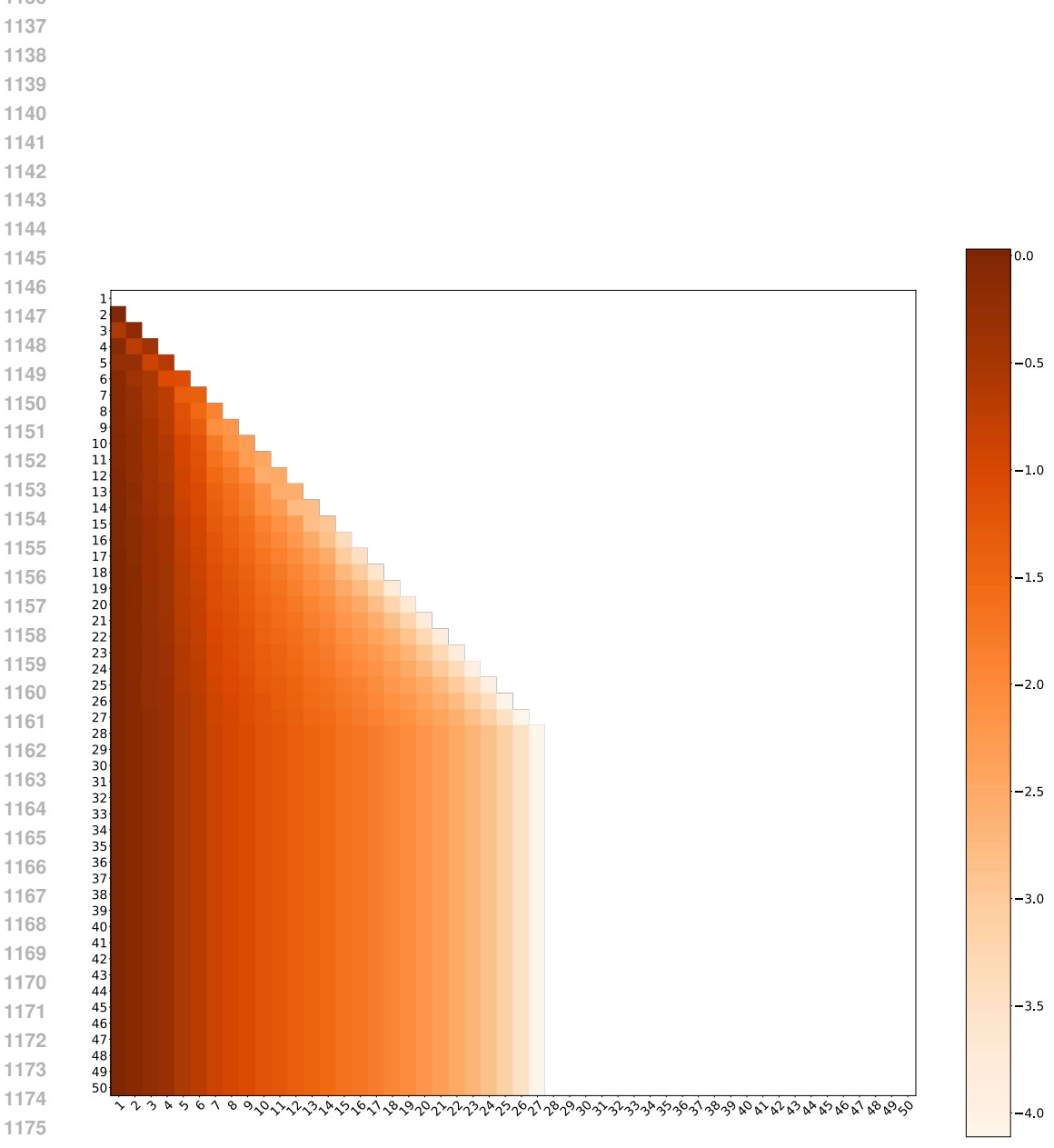

Figure 14: SRMT agent 2 heatmap of distances between memory states.

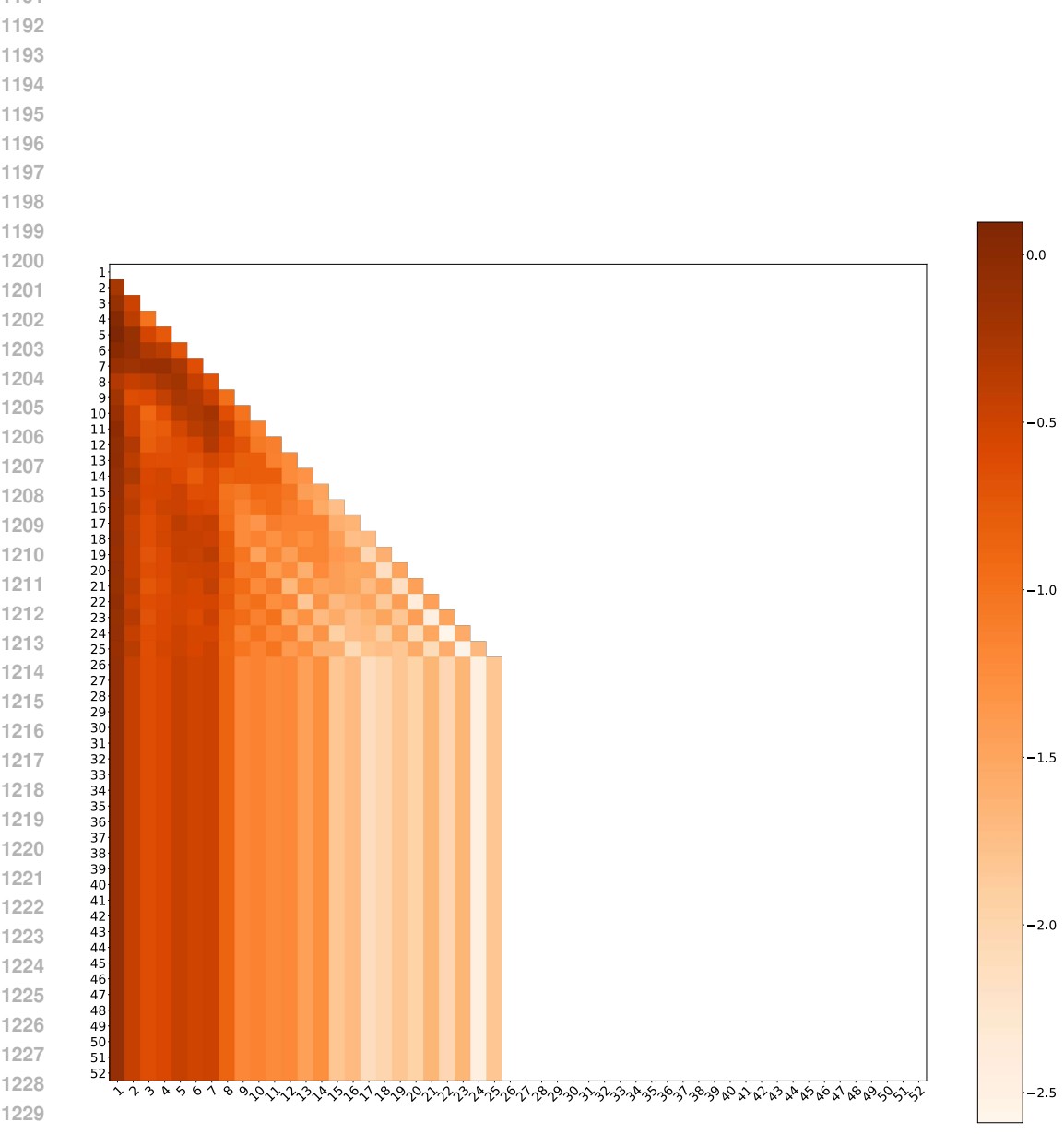

Figure 15: Agent 1 with RMT policy heatmap of distances between memory states.

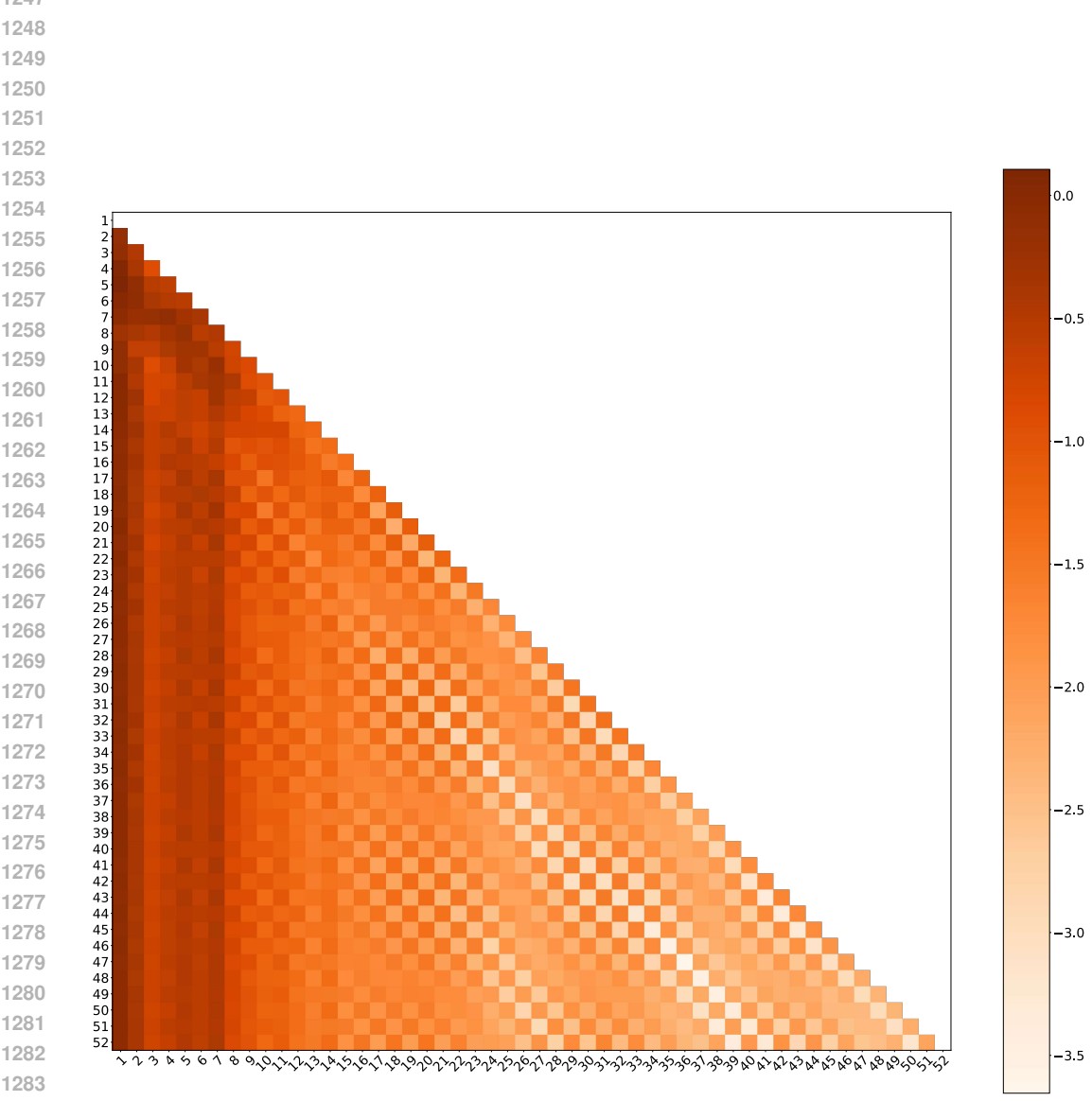

Figure 16: Agent 2 with RMT policy heatmap of distances between memory states.

