# OpenReview forum: "Shared Recurrent Memory Transformer for Multi-agent Lifelong Pathfinding"
_ICLR.cc/2026/Conference — Submitted to ICLR 2026_

### Official Review · Reviewer_hozZ · 2025-10-21

**Soundness:** 2
**Presentation:** 3
**Contribution:** 2
**Rating:** 4
**Confidence:** 2

**Summary:**

The paper proposes Shared Recurrent Memory Transformer (SRMT) for Partially Observable Multi-Agent Pathfinding (PO-MAPF) problem by enabling coordination of agents through unconstrained communication. Specifically, SRMT proposes a shared recurrent memory transformer to broadcast their recurrent memories to a global workspace with self-attention on individual memory and observation and cross-attention on shared agent memories. Results on six maze environments from the POGEMA benchmark show that SRMT outperforms various baselines.

**Strengths:**

1.	The motivation for using a global shared memory to help coordination in the PO-MAPF problem is clear.
2.	This paper is well-written and well-organized. The related works are well discussed.
3.	Performance of SRMT is very strong compared with various cooperative and memory baselines.

**Weaknesses:**

1.	The idea of a global shared memory with self-attention and cross-attention has limited novelty.
2.	When compared with advanced path-planning baselines such as RHCR, the performance of SRMT is not significant. It decreases the contribution of this work as centralized planning methods are already good especially SRMT also needs a global shared memory mechanism.
3.	SRMT is only tested with an algorithm PPO on POGEMA. Whether SRMT could be applied to other MARL algorithms is not clear.
4.	Other MARL algorithms with specially designed communication mechanisms are not compared.

**Questions:**

1.	Could SRMT be integrated into QPLEX or some other advanced MARL algorithms?
2.	How does SRMT compare with MARL algorithms with communication mechanism?
3.	Why there is still a large performance gap of SRMT when compared with RHCR in some POGEMA maps?

---

> ### Author Response · Authors · 2025-11-21
> **Author Response**
>
> **We thank the reviewer for their detailed and insightful feedback. We appreciate the recognition of the clear motivation of the global shared memory to help coordination in the PO-MAPF, a well-written and well-organized paper, and well-discussed related work, and very strong performance of SRMT compared to various cooperative and memory baselines. Below, we address each of your concerns and questions systematically. Please let us know if any issues remain!**
>
> **W1:**
> We agree that the primitives (self-attention, cross-attention) are standard. Our claim is not about inventing these components, but about how they are used for coordination in MARL. In MARL, recurrent states and memories are almost always per-agent (RRNN, RATE, ATM) and not used as a persistent shared workspace. SRMT is, to our knowledge, the first to make the memory jointly owned and updated by all agents across time, turning it into a communication channel rather than a private state.
>
> **W2 & Q3:**
> As it is discussed in the paper (see lines 420-422), RHCR is a centralized oracle that has full environment information and replans at every step, marking an unreachable theoretical upper bound for decentralized RL methods. However, centralized methods scale poorly with the growing number of agents as they require the full information about all the agents and the environment state at each time step to make a decision. SRMT is a decentralized method that performs the decision-making for each agent independently based only on its local observations, limited by the fixed-size observation window and the agent’s memory state. Decentralization allows efficient scaling of SRMT, while shared memory provides the mechanism for cooperation in the partial observability setting.
>
>
> **W3 & Q1:**
> SRMT is architecture-agnostic and can indeed be integrated into value-based MARL algorithms like QPLEX, QMIX, and others. The shared memory mechanism operates as a coordination module that enriches the joint value function decomposition. This is an interesting direction for future work; however, it is out of the scope of the current study.
>
> **W4 & Q2:**
> MAMBA presents a communicative MARL baseline in our experimental evaluations, as it has a specially designed communication block used by the agents to predict their current state feature vectors from stochastic states and actions from previous steps.
>
> On the Bottleneck task, MAMBA completely fails across all rewards, compared to SRMT (see Table 2 of the main paper).
> In Lifelong MAPF, SRMT outperforms MAMBA in average and out-of-distribution performance, generalizing to the unseen types of maps and a growing number of agents. Also, in cooperation metrics, SRMT outperforms MAMBA.

---

> > ### Comment · Reviewer_hozZ · 2025-11-28
> > **Thanks to the authors for the response**
> >
> > Thanks to the authors for the response. I would like to maintain my score. Some of my concerns (such as novelty compared to  UPDeT [1] with a global memory, applying SRMT to other MARL algorithms, comparing with more MARL approaches with communication, and the close performance compared to advanced path-planning baselines) are not fully addressed.
> >
> > References
> >
> > [1] Hu, Siyi, et al. "Updet: Universal multi-agent reinforcement learning via policy decoupling with transformers." arXiv preprint arXiv:2101.08001 (2021).

---

> > > ### Author Response · Authors · 2025-11-29
> > > **Response to comment, part 1.**
> > >
> > > We appreciate the provided reference. We are aware of this work and left it out of the related works section to avoid misleading the reader, as the UPDeT paper's objective is fundamentally different from the one presented in our paper.
> > >
> > > UPDeT aims to overcome the MARL transfer learning limitations, such as different observation and action configurations, and applies a transformer-based model to decouple the policy distribution from the intertwined input observation. The module referred to as “global hidden state” in the UPDeT comprises a hidden vector that tracks agent’s **individual** observation history and is used by each agent independently (similar to single-agent RL memory presented in RRNN, ATM, RATE). Such vector usage is motivated by the statement that “POMDP policy is highly dependent on the historical information” (see Section 3.2 Eq. 2 UPDeT paper). In contrast, SRMT’s shared memory provides each agent with information about **all** agents in the system and functions as an inter-agent networking channel for coordination rather than just historical information storage for general POMDP policy functioning.
> > >
> > > It is worth noticing that this terminology used in UPDeT may be somewhat confusing when contextualized within MARL communication literature.
> > >
> > > As UPDeT's focus is on the MARL transfer learning, it doesn't address the main challenges in MAPF, e.g., multi-agent coordination or collision avoidance. Thus, referencing UPDeT in the context of our paper can mislead the reader.
> > >
> > > We hope this discussion will clarify your question and we look forward to adding this clarification in the revised version of the paper.
> > >
> > > **We already compare SRMT with the highest performing MARL approaches with communication based on POGEMA benchmark evaluations.**
> > >
> > > Our evaluation includes comparisons with CTDE MARL approaches: cooperative QPLEX and communicative MAMBA. We selected these baselines based on the POGEMA benchmark [6] results: QPLEX along with RHCR, Follower, and MATS-LP are the highest performing methods in Lifelong MAPF task. MAMBA, also used in the POGEMA benchmark paper, presents a MARL communication method featured in Lifelong task evaluations. In our experiments, we demonstrate that SRMT significantly outperformed these methods across all Bottleneck environments and POGEMA benchmark scenarios with the exception of the relative runtime (Scalability) and relative local agent density (Congestion score), where it remained competitive.
> > >
> > > **We use advanced path-planning baselines as the evaluation upper bound.**
> > >
> > > While Figure 5 shows RHCR’s 100% in Performance, RHCR is a centralized oracle that knows the full map and re-plans every step, so its 100% score marks a theoretical upper bound that decentralized RL methods cannot reach. Nevertheless, SRMT outperforms RHCR in congestion management, proving the SRMT shared memory mechanism importance for solving the MAPF coordination challenge. Also, from the practical applicability point of view, the full observability requirement of RHCR makes it non-scalable to larger environments and agent populations. In contrast, SRMT is a decentralized approach working in partial observability setting. This ensures the method’s scalability and efficiency with various environment and agent population sizes.
> > >
> > > Decentralized methods for advanced path planning (Follower, MATS-LP) also serve as LMAPF performance upper bound as they combine tailored path search algorithms with policy training, while SRMT only employs the A* search.
> > >
> > > Follower’s heuristic search algorithm requires for each agent to know the **whole map's** obstacles configuration to calculate paths that would evenly disperse the agents over the map. While SRMT works only with the information provided in the **agent observations**, significantly decreasing the amount of required information about the environment. From the practical point of view, SRMT simulates a much more realistic scenario for decision-making rationale, where an agent knows only the already visited part of the map and that was accessible only through its field-of-view. Follower knows the locations of obstacles of the whole map at each time step.
> > >
> > > MATS-LP planner applies multi-agent neural MCTS that for each agent simulates future scenarios by building a search tree and chooses the best possible action at each time step. This procedure requires to calculate all possible actions for the agent itself and nearby agents, resulting in substantial decision time overhead compared both to Follower and SRMT. As a result, MATS-LP has the second lowest Scalability value (see Figure 5) which harms its practical applicability in large environments. Also, MATS-LP relies heavily on the specifically designed dense reward signal, while SRMT demonstrates superior performance with both sparse and dense reward signals (see Figure 4), providing a more generalizable solution.

---

> > > > ### Author Response · Authors · 2025-11-29
> > > > **Response to comment, part 2.**
> > > >
> > > > References:
> > > >
> > > > [1] Hu, S., Zhu, F., Chang, X., & Liang, X. (2021). UPDeT: Universal Multi-agent RL via Policy Decoupling with Transformers. International Conference on Learning Representations. https://openreview.net/forum?id=v9c7hr9ADKx
> > > >
> > > > [2] Wen, Muning, et al. "Multi-agent reinforcement learning is a sequence modeling problem." Advances in Neural Information Processing Systems 35 (2022): 16509-16521.
> > > >
> > > > [3] Zhao, Chengzhang, et al. "ACUTE: Attentional Communication Framework for Multi-Agent Reinforcement Learning in Partially Communicable Scenarios." Electronics 11.24 (2022): 4204.
> > > >
> > > > [4] Khan, Muhammad Junaid, Syed Hammad Ahmed, and Gita Sukthankar. "Transformer-based value function decomposition for cooperative multi-agent reinforcement learning in starcraft." Proceedings of the AAAI Conference on Artificial Intelligence and Interactive Digital Entertainment. Vol. 18. No. 1. 2022.
> > > >
> > > > [5] Zhao, Qingpeng, et al. "Boosting value decomposition via unit-wise attentive state representation for cooperative multi-agent reinforcement learning." arXiv preprint arXiv:2305.07182 (2023).
> > > >
> > > > [6] Skrynnik, A., Andreychuk, A., Borzilov, A., Chernyavskiy, A., Yakovlev, K., & Panov, A. (2024). Pogema: A benchmark platform for cooperative multi-agent pathfinding. arXiv preprint arXiv:2407.14931.

---

### Official Review · Reviewer_31Ta · 2025-10-27

**Soundness:** 3
**Presentation:** 3
**Contribution:** 3
**Rating:** 8
**Confidence:** 4

**Summary:**

The paper proposes the Shared Recurrent Memory Transformer, SRMT, an architecture for decentralized multi agent reinforcement learning in partially observable multi agent pathfinding. Instead of relying on a fixed, hand designed communication protocol or on centralized training that does not scale, SRMT gives each agent a recurrent working memory state, broadcasts those memory states into a shared global memory workspace, and then lets every agent query that shared memory via cross attention at every timestep. The idea is that agents can implicitly coordinate by aligning their internal memory representations, rather than by passing explicit structured messages. The model is trained end to end with PPO in a fully decentralized setting, the same policy weights are shared across agents, and no centralized controller is used at execution time. The paper evaluates on two settings, a minimal two agent Bottleneck task that requires negotiation to get through a one cell corridor without deadlock, and large scale partially observable multi agent pathfinding (both classical and lifelong) using the POGEMA benchmark, including environments with up to hundreds of agents. On Bottleneck, SRMT solves the task even with sparse rewards, and generalizes to corridor lengths up to 1000 cells despite only training on corridors of length 3 to 30. On POGEMA, SRMT is competitive with or ahead of centralized training methods and memory based baselines, and scales to 128 plus agents with reasonable throughput, matching or beating communication heavy baselines in coordination metrics and in congestion handling. The paper also presents ablations which suggest that shared memory is essential in the hardest settings, where agents must negotiate under weak or delayed feedback, and that the learned shared memory actually encodes inter agent coordination state, because the cosine distance between two agents’ memories shrinks before they meet in the corridor and stays aligned while they resolve who yields.

**Strengths:**

SRMT is a clean, well motivated architectural idea. Instead of engineering what messages agents should send, or imposing a fixed-bandwidth channel, the paper proposes a shared workspace that all agents write to and read from through attention. This is conceptually similar to a blackboard system in classical distributed AI, but implemented as differentiable recurrent memory, and updated every timestep inside a transformer block with self attention on each agent’s local history and cross attention to the shared workspace. This lets the method keep decentralized execution, which is crucial in large swarms, while still allowing high bandwidth coordination, which typical CTDE or bandwidth limited communication schemes struggle with. The Bottleneck task is a nice diagnostic experiment. It isolates negotiation, forces agents to decide who yields in a single lane corridor, and stresses partial observability when the corridor is longer than the view range. SRMT achieves perfect cooperative success rate in both directional reward and sparse reward settings, where baselines like MAMBA, QPLEX, ATM, and RRNN either fail completely or only solve easier reward structures, and SRMT keeps working even when the reward gives no dense guidance along the way. The generalization test is also impressive, SRMT trained only on corridors of length 3 to 30 successfully coordinates in corridors up to length 1000 at test time, maintaining near perfect success at lengths up to 400 in the sparse reward case, and remaining the top performer in moving negative reward all the way out to 1000, which is strong evidence that it learned a transferable negotiation routine, not memorized scripts for a particular map size. The memory analysis is compelling. The paper shows that cosine distance between two agents’ memory vectors shrinks as they approach each other, stays low while they pass in the corridor, and then increases again once they are free to pursue their own goals, so the shared memory is not just a dump of observations, it is aligned with “we are currently coordinating” versus “we are independent” phases. On POGEMA, which is widely used for partially observable multi agent pathfinding, SRMT remains competitive or better than CTDE based learners like QPLEX, memory baselines like RATE and RRNN, and even planning based methods on some maps, while requiring no domain specific heuristics. The paper also pushes into lifelong multi agent pathfinding, where agents keep getting new goals instead of terminating, and shows that SRMT can be scaled to dozens of agents, then mixed population training improves further, and that adding a heuristic path decider improves congestion handling. Finally, the authors include system level measurements, steps per second and GPU memory footprint of SRMT with and without shared memory, and show roughly linear scaling in agent count in terms of runtime per step, which is important for claims about practical scalability.

**Weaknesses:**

The approach still assumes an effectively unconstrained high bandwidth shared memory bus, which is conceptually decentralized, but physically looks like broadcast of latent state to everyone at every timestep. In other words, the method offloads coordination into a global attention accessible structure, which in many practical robot or wireless settings would itself be a bottleneck, so while the paper argues that SRMT maintains decentralized execution, it is really assuming reliable, synchronous, instantaneous global memory sharing, which is a strong assumption in large fleets. The paper acknowledges this only briefly in the limitations, and it would be useful to quantify how performance degrades if access to the shared memory is lossy, delayed, or spatially restricted. Almost all experiments assume homogeneous agents with identical policies and identical capabilities. This is a hard setting in terms of emergent symmetry breaking, and the paper is right to point that out, but it is also the easiest setting for weight sharing and for a shared latent memory workspace, because everyone writes embeddings in the same format. It is unclear whether SRMT would remain stable and interpretable if different agents had different roles or actuation constraints, or if some had different observation ranges. Ablations are strong for Bottleneck, but thinner for the large scale POGEMA case. For Bottleneck, the paper shows that removing the memory head, history, or shared memory hurts badly, which supports the claim that shared recurrent memory is really doing the work, not just generic recurrence. For lifelong POGEMA, the ablations are mostly variants of SRMT with or without heuristic path decider and with different training population sizes, but do not fully isolate how much of the gain comes from shared memory in the many agent regime relative to, say, just a large transformer with private memory and a good heuristic injector. Evaluation is broad in terms of environments, but still somewhat narrow in terms of baselines. The paper compares with CTDE style QPLEX and cooperative MARL methods, and with path planners like RHCR and Follower, but I did not see explicit comparisons to more recent transformer based communication architectures like SCRIMP in the same partially observable settings, or to implicit communication via learned messaging under bandwidth constraints. This leaves a small gap in the claim that SRMT is strictly better than communication learning, because the competitors either assume centralized training or use memory without sharing. Some of the large scale throughput plots are difficult to interpret without additional statistics. For example, SRMT plus heuristic planning beats others on congestion and throughput in warehouse style maps, but MAMBA still produces a higher raw throughput in that single warehouse configuration. The paper attributes this to reduced diversity of layouts in that evaluation, but a more explicit discussion of failure modes would help, for example, where SRMT’s coordination logic breaks down compared to a heuristic planner, or situations where shared memory entrains everyone into the same lane and causes traffic jams. Finally, the shared memory workspace looks a lot like a differentiable blackboard, but there is not yet a learned discipline about what to write there. It would be interesting to know if the model ever writes misleading or stale information that hurts others, or if it ever “lies” to negotiate, for example stalls another agent on purpose to grab priority. These behaviors are important for safe deployment in real multi robot systems, and are not analyzed.

**Questions:**

How realistic is the shared memory assumption in real swarms. Concretely, do you assume every agent can broadcast its memory vector to every other agent at every step with no delay and no packet loss. Have you tried injecting communication delay or partial observability into the shared memory itself, for example only nearby agents’ memories are visible, and if so, how badly does performance drop on Bottleneck and POGEMA. Can you report a fairness controlled comparison on POGEMA against a bandwidth limited communication baseline, for example SCRIMP style transformer communication with a fixed message budget, so we can see if SRMT still wins when the other method is not artificially bottlenecked. In lifelong multi agent pathfinding on large maps, what is the failure mode when SRMT fails. Is it deadlock, local congestion, oscillation, or just extremely long detours. A short qualitative analysis would help clarify what the shared memory is and is not learning. In Figure 3, you show that cosine distance between agent memories decreases as agents approach each other and stays low during corridor negotiation. Can you expand this analysis to more than two agents, for example, in POGEMA warehouse style congestion, do clusters of agents locally align their memory states in the same way, does the shared memory reflect global traffic patterns like “I am blocking aisle 5, you reroute”. How sensitive is SRMT to the observation patch size and the history window h. You ablate history length, and performance collapses without history on Bottleneck under hard rewards, which is informative. Could you also report sensitivity curves for observation range in POGEMA or for the memory dimensionality in SRMT. Finally, you report steps per second and memory usage up to 1024 agents on an A100. Can you comment on whether the cross attention to the shared memory is the dominant runtime cost, and whether you explored sparse attention or locality restricted attention, which may be important for scaling beyond 1k agents or for deployment on cheaper hardware.

---

> ### Author Response · Authors · 2025-11-21
> **Author Response, part 1.**
>
> **We sincerely thank the reviewer for their detailed and constructive feedback. We greatly appreciate the positive scores and positive comments on the paper’s clean, well-motivated architectural idea, and the nice diagnostic experiment of the Bottleneck task. The reviewer also acknowledges the impressive generalization test, compelling memory analysis, competitiveness of SRMT on POGEMA, and the practical scalability of the method. Below, we address each of your concerns and questions systematically.**
>
>
> **W1 & Q1:**
> Indeed, the  Appendix B Limitations section of our paper reveals that we consider an abstraction of the MAPF problem that is very actively studied in the community. As the primary focus of this work is the decision-making mechanism improvement, for fair unified comparison with other approaches, we assume that the agents execute actions accurately and that their moves are synchronized, which allows at every time step for every agent to pool its memory state into the shared memory that is further broadcast to all agents with no delay and no packet loss.
>
> We acknowledge that in realistic multi-agent robotic system applications, communication is affected by delays, spatial restrictions, and packet loss. The imperfection is also inherent in robots themselves, as they rarely execute the prescribed actions or trajectories perfectly. The assumptions on communication synchronicity made in our paper are also made in many other works in the field, including MAMBA (Egorov et al., 2022), Follower (Skrynnik et al., 2024), and MATS-LP  (Skrynnik et al., 2023), with whom we compare.
>
> The comparison of SRMT to a bandwidth-limited communication baseline is already presented in the paper - MAMBA is a method with discrete message communication, specifically designed to address the limited communication channel bandwidth constraint. Our experimental evaluations show that SRMT significantly outperforms MAMBA in various environments, both in and out of training distribution, and also maintains superior performance in the case of the growing number of agents (from 64 to 256 agents as reported in Figure 12 of Appendix D).
>
> **W2:**
> SRMT operates in the hidden space of the model. The memory vectors and SRMT processing are not affected by the differences in agent roles or actuation constraints because SRMT takes as input only the encoded representations with dimensionality equal to the hidden space one, and outputs the representations of hidden space dimensionality. These output representations can further be easily decoded into the required action space with any dedicated decoder. The modularity of SRMT is one of its key features: performing only in the hidden space, SRMT can be integrated with any desired encoders and decoders as soon as they can perform respective vector representations of hidden space dimensionality. This makes SRMT a general implementation-friendly solution for the injection of the shared memory cooperation mechanism.
>
> **W3:**
> We use the Bottleneck environment for our ablation study that tests architectural factors in isolation (shared memory, private memory, history, and memory head), as this environment provides a clear, minimalistic, and fully controllable scenario for the fair comparison of ablated variants of SRMT. This study clearly demonstrates that the proposed shared memory mechanism is the dominant factor in performance improvement compared to, for example, attention mechanism, individual memory, or generic recurrence. We use a Lifelong MAPF task from the POGEMA benchmark for a large-scale evaluation, a long-term generalization testbed to determine if the proposed approach is beneficial when scaled to many agents and harder tasks. The variants of SRMT trained with a heuristic path planner or with a variable number of agents are not in fact, ablations as they are not supposed to remove any part of the architecture but to add an advanced planning search algorithm or broaden the size of the agent’s populations used for training.
>
> Even in POGEMA, the gains of SRMT are measured against strong baselines without shared memory, so the improvement cannot be attributed solely to “a large transformer + heuristic injector”.
>
> **W4:**
> For Lifelong MAPF evaluations, we selected as baselines the methods that are top-performing in the lifelong setting of the POGEMA benchmark (RHCR, Follower, MATS-LP, QPLEX) to demonstrate the SRMT effectiveness compared to the best-performing learnable and non-learnable methods for the given task. Also, we used MAMBA as a learnable implicit transformer-based communication mechanism that operates with discrete messages to account for message channel bandwidth limitations. In our experiments, both on classical and lifelong MAPF, we demonstrated that SRMT has superior performance and scalability (both with the environment size and the number of agents) over MAMBA. This baseline exactly addresses your concern.

---

> > ### Author Response · Authors · 2025-11-21
> > **Author Response, part 2.**
> >
> > **W5:**
> > The heuristic search algorithm implemented in the Follower aims to evenly disperse the agents over the map by penalizing the deadlock-prone paths. To calculate such paths, the planner for each agent requires knowing the whole map's obstacles configuration, while SRMT is based only on the information provided in the agent observations with a small fixed field-of-view. Thus, SRMT operates with significantly less information about the environment.
> >
> > On the other hand, SRMT simulates a much more realistic scenario for decision-making rationale, where an agent knows only the already visited part of the map and that was accessible only through its field-of-view, compared to Follower, which knows the locations of obstacles of the whole map at each time step.
> >
> > **W6 & Q2:**
> > We appreciate the experimental suggestions of the reviewer. Our goal was to present a novel coordination approach based on the shared memory mechanism and to prove its performance and scalability effectiveness in both classical and lifelong MAPF settings through a thorough experimental evaluation and ablation. exploring the model failure modes in a truly interesting direction for future research.
> >
> > **Q3:**
> > In Table 4 of the main paper, we present time and memory consumption for the model with and without shared memory, to test how shared memory addition affects the model's runtime cost. It can be seen from the results that SRMT without shared memory has slightly higher values of the steps per second metric than the original SRMT with shared memory, showing that shared memory incorporation influences the runtime values.
> > We have not explored sparse or locality-restricted attention for scaling beyond 1k agents in our experiments, but will consider it as a future line of work. We appreciate the suggestions!

---

> > > ### Comment · Reviewer_31Ta · 2025-11-27
> > >
> > > 1. A lot of researchers are now shifting toward using LLMs to help with MARL communication. However, I still see strong value in the more traditional direction of improving the structure of multi-agent communication itself like this paper did. The shared memory transformer is a solid idea, the use of the attention mechanism makes it much more practical for MA-Comm researchers to build new algorithms on top of it, rather than just dropping an LLM module into the architecture. The comparison with CTDE is also helpful, this work is basically trying to find a smarter, more flexible way to do CTDE through a shared memory pool, which I really like.
> > > 2. I know some reviewers are concerned about novelty, but from my experience working in multi-agent communication for years, even extending an existing architecture and making it actually work in a multi-agent setting is already very hard. Getting stable improvements, adapting the design to a multi-agent setting, and making it scale in MA environments takes a lot of insight and engineering. Novelty in this area often comes from these adaptations, not from inventing something completely new from scratch.
> > > 3. I will keep my score. Thanks to the authors for answering and addressing my questions thoroughly.

---

> > > > ### Author Response · Authors · 2025-11-28
> > > >
> > > > Thank you for your thoughtful review and for recognizing the value of our structural, attention-based approach to multi-agent communication as a more transparent and extensible alternative to LLM-based methods. We deeply appreciate your understanding that successful adaptation to multi-agent settings requires substantial insight and engineering, and we're grateful for your continued support of our work!

---

### Official Review · Reviewer_MLNw · 2025-10-31

**Soundness:** 2
**Presentation:** 3
**Contribution:** 2
**Rating:** 4
**Confidence:** 3

**Summary:**

This paper addresses the Partially Observable Multi-Agent Path Finding (PO-MAPF) problem by introducing a new architecture called Shared Recurrent Memory Transformer (SRMT). Instead of explicit message passing or centralized training, SRMT allows all agents to read and write to a shared recurrent memory via attention, effectively forming a global workspace for implicit communication. Each agent encodes its own history and observation through self-attention, interacts with the shared memory through cross-attention, and updates its own state recurrently. The method is evaluated on the Bottleneck coordination task and POGEMA environments, demonstrating improved coordination and scalability under sparse reward and partially observable settings.

**Strengths:**

1. Well-motivated problem setting. The paper clearly articulates the challenges of decentralized coordination under partial observability and sparse rewards, which are highly relevant to real-world multi-robot and swarm scenarios.

2. Conceptually simple, implementation-friendly. The shared-memory attention mechanism is intuitive, modular, and can be easily plugged into existing transformer-based MARL pipelines.

3. Strong empirical validation. Experiments on the Bottleneck and POGEMA benchmarks are systematic, covering different reward densities, map scales, and agent counts.

4. Good interpretability. The visualization of memory alignment between agents (e.g., cosine similarity peaking near encounters) provides evidence that the shared workspace learns meaningful coordination semantics.

**Weaknesses:**

1. Limited novelty. SRMT mainly reconfigures known ideas (shared latent space + recurrent transformer) rather than introducing a fundamentally new learning principle.

2. Restricted task diversity. All experiments are in grid-world pathfinding domains (POGEMA and its variants). There are no tests on heterogeneous agents, dynamic obstacles, or more complex continuous-control tasks, which weakens claims of generality.

3. Lack of theoretical or conceptual depth. The paper positions SRMT as a general coordination mechanism, but provides no formal analysis or ablation beyond empirical metrics to support the claimed emergence of “implicit communication.”

**Questions:**

1. Since every agent attends to the shared memory of all agents, how does the method scale computationally as the number of agents increases? Have you tested SRMT in settings with hundreds of agents?

2. How generalizable is SRMT to heterogeneous or continuous-control environments?

3. Are the improvements over baselines primarily due to the shared-memory mechanism or to larger model capacity and richer representations?

4. The “unconstrained communication” assumption may not hold in realistic multi-robot systems. How would SRMT perform under communication delay or bandwidth limits?

---

> ### Author Response · Authors · 2025-11-21
> **Author Response**
>
> **We thank the reviewer for their constructive feedback. We appreciate the acknowledgment of SRMT’s well-motivated problem setting, the intuitive and implementation-friendly shared-memory attention mechanism, the strong empirical validation of SRMT, and the good interpretability of the shared workspace. We address your questions below.**
>
> **W1:**
> We appreciate the reviewer’s concern and agree that SRMT is built from existing components (transformers, memory, shared latent representations). Our contribution is not a new learning principle, but a new architectural mechanism for coordination in multi-agent RL.
>
> To our knowledge, prior MARL work with memory (RRNN, RATE, ATM) maintains private recurrent memories per agent, which are never used as a global communication medium. Conversely, communication-based MARL methods (e.g., MAMBA, QPLEX, SCRIMP) implement explicit message passing or centralized value factorization. However, they do not provide a shared "workspace" that is jointly written and read by all agents at every step. In SRMT, each agent uses cross-attention from its local history to this shared memory and updates it via a learned read-write rule, learning what to store for others.
>
> We have added the explicit clarification of the architectural novelty at the end of the introductory section of the revised manuscript.
>
> **W2:**
> The choice to focus our evaluation on grid-world MAPF allowed us to conduct a thorough, controlled, and unified evaluation of SRMT's core mechanisms — agents’ individual memory and the shared memory space for communication — in a well-understood environment with extensive generalization tests. We believe the underlying transformer architecture and the memory mechanisms of SRMT are general and not specific to the considered action and observation spaces, suggesting strong applicability to other complex multi-agent domains. Extending our evaluation to different benchmarks is the next step for future work, which is highlighted in the Limitations section in Appendix B.
>
> **W3:**
> Could you please specify what kind of formal analysis or theory you mean?
>
> To assess the shared memory coordination efficiency, as a part of the ablation study, we compared SRMT with a memory-less transformer, referred to as Attention, in Table 3 of the paper, and demonstrated the significant performance drop in Attention model compared to SRMT.
>
> **Q1:** In Table 4 of the main paper, we provide the time and memory consumption of SRMT with and without shared memory for populations of up to 1024 agents. Shared memory has no significant effect on inference time, maintaining close to linear scaling with the number of agents. However, a larger memory amount is required for 512 agents and more for SRMT compared to the model without shared memory.
>
>
> **Q2:**
> The proposed shared memory mechanism is general and does not require any change to be used in heterogeneous or continuous-control environments, as it operates in the hidden space of the model. So SRMT can be seamlessly integrated into the heterogeneous tasks as soon as agents’ observations or any other input features can be encoded into vector representations of hidden space dimensionality.
>
> **Q3:**
> To distinguish the contribution of the shared memory, we conduct the ablation study (Table 3 of the main paper), where SRMT is compared to the memory-less transformer of the same size, referred to as the Attention model. In Moving Negative and Sparse rewards, the Attention model shows significant performance drops compared to SRMT: the Attention model has 0.4 and 0.7 average CSR, for respective rewards, compared to 0.8 and 1.0 CSR for SRMT.
>
>
> **Q4:**
> In the Appendix B Limitations section of the paper, we disclose that we consider agents that execute actions accurately and have synchronized moves, allowing for unconstrained communication.
>
> We acknowledge that in real-world robotic systems in the real-world robotic systems communication is not perfect and delays and bandwidth limits are the norm. And robots themselves – they rarely execute the prescribed actions or trajectories perfectly. In this paper, however, we target not the realistic multi-robot systems applications but the abstraction of the actively studied problem - MAPF. It is worth noting that numerous real-world multi-robot system challenges are raised in this problem abstraction. In this work, we demonstrate how shared memory can be added to a decentralized, learnable MAPF solver to enhance the coordination and overall performance of the multi-agent system.

---

### Official Review · Reviewer_Vgu5 · 2025-11-03

**Soundness:** 2
**Presentation:** 2
**Contribution:** 2
**Rating:** 4
**Confidence:** 3

**Summary:**

This paper proposes a shared recurrent memory transformer (SRMT) architecture for coordination in multi-agent systems. The paper particularly focuses on the problem of cooperative path planning, in which each agent aims to find a path to its goal location, without colliding or interfering with other agents. The intuition of the SRMT is that its cross attention weights, when applied to agents’ memories, allow for implicit coordination between agents. Unlike centralized training and decentralized execution (CTDE) methods, however, SRMT allows each agent to make its action decisions individually, even in the training phase, leading to greater scalability to many agents. Experiments on a variety of pathfinding environments show that SRMT outperforms other CTDE baselines, as well as decentralized baselines that utilize memory.

**Strengths:**

+ Cooperative pathfinding is an important application scenario, e.g., for robotics applications, and it exemplifies many of the challenges in coordinating multi-agent systems.

+ The proposed method scales well to a large number of agents, supporting up to 128 agents in simulations as shown in Table 4’s results on inference efficiency. Intuitively, the cross-attention mechanism in the SRMT should be easier to train than CTDE methods that require training over combinations of agent actions.

**Weaknesses:**

--The intuition and the technical challenges behind using the SRMT for agent cooperation are not explained well in the paper, and thus it’s difficult to fully appreciate SRMT’s novelty or technical merit. The SRMT architecture seems to be a standard transformer.

--It’s also not clear how the SRMT would have access to memories from other agents. Wouldn’t agents need to share these memory states in order to pass them into the SRMT? Does each agent maintain its own trained SRMT, or is a single SRMT model maintained at a central server? How is the SRMT combined with the PPO framework? Without these details, it is difficult to appreciate the proposed method.

--In the experiments, the paper utilizes three different reward functions in order to showcase different aspects of cooperation. However, the choice of reward seems rather artificial as these functions are all applied to the same environments (and in general, one can choose any desired reward function in reinforcement learning, so long as it is observable). At a minimum, example settings in which each reward function would be appropriate should be given so that the rewards appear to be realistic instead of artificially constraining the evaluated methods.

**Questions:**

Please see also the weaknesses above.

1) What was the overall (not just per-step) training time for SRMT compared to MAMBA, and how does it scale with the number of agents?

2) It’s not clear how many agents were used to obtain the experiment results in Figure 2 and Table 1. Why is comparing SRMT with 64 agents a fair comparison to SRMT with a mix of 64 and 128 agents—wouldn’t a different number of agents change the environment entirely? Or can the number of agents present in the environment change over time? How many agents were used in the baselines (MAMBA, etc.)?

---

> ### Author Response · Authors · 2025-11-21
> **Author Response, part 1.**
>
> **We thank the reviewer for their constructive feedback. We appreciate the recognition of cooperative pathfinding as an important application scenario, the scalability of SRMT to a large number of agents, and the ease of SRMT training compared to CTDE methods. We address each concern systematically below and provide additional clarifications that we believe strengthen our contribution.**
>
>
> **W1:**
> SRMT, unlike a standard transformer applied to multi-agent settings, introduces the following key innovations:
> 1. Shared memory pool architecture: unlike the conventional transformers case, where each agent uses a transformer independently, SRMT maintains a distributed but globally accessible memory pool. Each agent writes to its dedicated memory slot while also being able to read from all other agents' slots via the cross-attention mechanism. This creates an implicit coordination channel without centralized decision-making.
> 2. Each agent's SRMT module attends to other agents' memory states containing information from previous timesteps. This ensures decentralized execution—agents never access current-step information from others, only historical context. This is fundamentally different from CTDE, where a central critic conditions on joint actions.
>
> One of the main technical challenges is the following: centralized methods fail to scale efficiently with the growing number of agents compared to decentralized approaches. However, decentralization in partially observable environments comes at a cost of no access to information about the locations and behavior of agents outside the tight observation window. This information is crucial for optimal solving of the path finding problem as agents have to avoid collisions and minimize congestion. SRMT presents a decentralized solution that efficiently scales with the size of the environment and the agent population size, and also provides agents with a coordination mechanism that allows for implicit negotiation and coordination required for optimal path finding.
>
> We appreciate the need for a better explanation in the paper and have added this discussion in the Appendix C of the updated version of the paper.
>
> **W2:**
> > How is the SRMT combined with the PPO framework?
>
> We use a standard decentralized PPO with a single actor-critic network shared between agents. The weights are shared between the actor and the critic. As shown in Figure 1, SRMT is incorporated in the policy network as an intermediate module, following the Spatial Encoder module.
> > Wouldn’t agents need to share these memory states in order to pass them into the SRMT?
>
> Each agent maintains its individual memory state that is passed as input to SRMT, along with the encoded observation, and is updated. After all agents update their individual memory states on the current step, these states are collected into a shared memory tensor. On the next time step, for each agent, SRMT processes a new encoded observation along with the current states of the individual and shared memory of the agent.
>
> We have added pseudocode in Appendix C to make the SRMT integration into PPO explicit.
>
> **W3:**
> You are correct that the reward design warrants better motivation. These functions simulate the following realistic multi-robot deployment scenarios:
> - Directional reward may be exemplified with automated baggage handling systems in the airport, where the only objective is bag delivery (reward=+1) and moving toward the gate (reward=+0.005) is slightly preferred to any other moving or staying in the same location (reward=0).
> - Moving Negative reward models the warehouse automation scenario when thousands of autonomous robots with battery limitations are fetching items in fulfillment centers, and their only goal is task completion(reward=+1). When the agent moves, its battery drains faster than when it is idle (reward for moving=-0.01, for staying=-0.005).
> - Sparse reward simulates the minimalistic setting for the warehouse robots where only task completion matters. This tests emergent coordination under sparse signals.
>
> We have added a "Reward Design Motivation" paragraph in Appendix D, connecting each function to real-world applications. While any reward is "choosable," these specifically test different coordination difficulty regimes—our contribution is showing SRMT adapts across them, whereas baselines succeed in only one.

---

> > ### Author Response · Authors · 2025-11-21
> > **Author Response, part 2.**
> >
> > **Q1:** We reveal the overall training time of SRMT and MAMBA in lines 471-472. In the Bottleneck task, SRMT training for 20M steps took 99 minutes, MAMBA training for 200k steps – 325 minutes.
> > In the table below, we provide the overall training time (in minutes) of SRMT (20M steps) and MAMBA (200k steps) on a large test map for the growing number of agents. For the train time test, we used a single NVIDIA H100 80GB with 128 Intel(R) Xeon(R) Platinum 8462Y+ CPU’s.
> >
> > | Agents | SRMT | MAMBA   |
> > |-|-|-|
> > | 32    | 100| 243 |
> > | 64    | 100| 316 |
> > | 128  | 96| 467 |
> > | 256  | 100| 750 |
> > | 512  | 100| 1353 |
> > | 1024| 101| 2823 |
> >
> > SRMT implementation is based on the very efficient PPO realization from the Sample Factory library, which provides a distributed training pipeline, resulting in the efficient overall training time. For the MAMBA test we used the original training pipeline implementation.
> >
> > **Q2:** Table 1 and Figure 2 do not provide any experimental results. Table 1 summarizes the baselines used in the study. Figure 2 illustrates the types of environments used for experimental evaluation.
> >
> > > Why is comparing SRMT with 64 agents a fair comparison to SRMT with a mix of 64 and 128 agents—wouldn’t a different number of agents change the environment entirely?
> >
> > The environment configuration of obstacles is fixed and independent of the agent population size. To evaluate how SRMT trained in scenarios with a smaller number of agents scales to growing agent populations, we trained SRMT with 64 agents and with a mixture of 64 and 128 agents, while evaluation was performed for scenarios with 64, 128, 192, and 256 agents. The detailed results are presented in Figure 12 of Appendix D.
> >
> > > Or can the number of agents present in the environment change over time?
> >
> > In Lifelong MAPF experiments, the number of agents present in the environment cannot change over time.
> >
> > > How many agents were used in the baselines (MAMBA, etc.)?
> >
> > The baselines in Lifelong MAPF experiments were trained on scenarios with 64 agents, as well as our main training configuration of SRMT for Lifelong MAPF.

---

### Meta-Review · Area_Chair_eVfY · 2026-01-02

**Summary:**

This paper proposes the Shared Recurrent Memory Transformer (SRMT), a transformer-based architecture for decentralized multi-agent reinforcement learning (MARL), targeting the Partially Observable Multi-Agent Pathfinding (PO-MAPF) problem. SRMT introduces a shared memory mechanism for implicit communication by allowing agents to read and write to a global memory space via attention.

While the paper addresses a timely and important problem in MARL and presents promising empirical results on coordination tasks like Bottleneck and POGEMA, several reviewers raised critical concerns regarding:

**Limited novelty**: Multiple reviewers (Vgu5, MLNw, hozZ) argued that SRMT is a reconfiguration of known mechanisms (transformers, shared latent memory) rather than introducing fundamentally new learning principles.\
**Lack of theoretical depth or analysis**: Reviewers noted the absence of formal analysis, theoretical insights, or principled understanding of how the shared memory contributes to coordination beyond empirical metrics (MLNw, hozZ).\
**Assumptions around communication**: SRMT assumes unconstrained, synchronous, global communication via shared memory, which is unrealistic for practical multi-robot systems. The paper only briefly discusses this limitation (31Ta, hozZ).\
**Restricted evaluation domains**: The method is only tested in grid-based navigation environments, with no validation on continuous control or heterogeneous agents (MLNw, 31Ta).\
**Baseline comparisons**: Although the authors compare against several CTDE and communication-based methods (e.g., MAMBA, QPLEX), some reviewers (hozZ) found the choice of baselines insufficient and noted the lack of comparison to recent transformer-based communication methods under bandwidth constraints (e.g., SCRIMP).\
**Scalability concerns**: While SRMT is evaluated with up to 1024 agents, some reviewers questioned whether the cross-attention mechanism to shared memory is scalable in practice, especially without sparse or locality-restricted attention (31Ta).\

These concerns, especially regarding novelty, realism of assumptions (especially on unconstrained communication), and breadth of evaluation, weighed heavily in the overall assessment.

**Recommendation: Reject**\
While the paper presents a well-engineered and clean architecture that performs well empirically in specific coordination tasks, the limited novelty, unrealistic assumptions, and restricted evaluation scope limit its impact. The core idea—using attention over shared memory—is interesting but not fundamentally new. The paper would benefit from broader evaluations (e.g., in continuous control), more realistic communication modeling, and clearer positioning relative to recent transformer-based MARL methods.

**Reviewer Concerns:**

**Addressed Concerns**\
The authors provided detailed and thoughtful rebuttals. Specific issues that were adequately addressed include:

- Clarified the architectural differences between SRMT and prior MARL memory models (e.g., RRNN, UPDeT), emphasizing the use of memory as a coordination medium rather than private state.\
- Motivated the choice of reward functions and connected them to real-world scenarios (Reviewer Vgu5).\
- Provided training time comparisons and scalability data for SRMT vs. MAMBA (Appendix D).\
- Explained how SRMT could generalize to heterogeneous agents and continuous control, citing its modularity and operation in hidden space.\
- Justified comparison against MAMBA as a communication baseline under message bandwidth constraints.

**Outstanding Concerns**\
Despite the rebuttal, key concerns remain unresolved:

- ***Novelty remains limited***: While the architectural design is clean and practically useful, the reviewers consistently noted that the core ideas (cross-attention + shared memory) are not fundamentally novel, and the work lacks conceptual or theoretical breakthroughs.\
- ***Communication assumptions are unrealistic***: SRMT assumes perfect, instantaneous message sharing, which undermines claims of decentralized realism. No experiments were conducted with delayed, lossy, or bandwidth-limited communication.\
- ***Insufficient generalization***: The method is only tested in grid-world navigation. No evidence is provided that SRMT can perform well in diverse, real-world domains such as robotic control, mixed-agent teams, or tasks with dynamic or continuous environments.\
- ***Baselines remain limited***: While the authors compare to strong methods, the lack of comparison to other recent transformer-based decentralized communication methods weakens the claim that SRMT advances the state of the art in MARL communication.

**Reviewer Scores:**

**Reviewer Vgu5 (Initial Score: 4)**: Raised concerns about novelty, clarity of SRMT’s implementation, and reward design. The rebuttal addressed reward motivation and implementation details, but novelty concerns likely remain. Predicted Final Score: 4 (unchanged)

**Reviewer MLNw (Initial Score: 4)**: Appreciated the motivation and modularity but questioned novelty, evaluation diversity, and theoretical depth. Rebuttal partially addressed these, but core concerns likely persist. Predicted Final Score: 4 (unchanged)

**Reviewer 31Ta (Initial Score: 8)**: Strongly positive and maintained score after discussion. Recognized practical utility and engineering effort in adapting the architecture to MARL. Final Score: 8 (unchanged)

**Reviewer hozZ (Initial Score: 4)**: Questioned novelty, baseline choice, and generality. Acknowledged the response but maintained concerns, especially regarding comparison to UPDeT and communication assumptions. Final Score: 4 (confirmed after discussion)

---

### Decision · Program_Chairs · 2026-01-26

Reject